# Stable species boundaries despite ten million years of hybridization in tropical eels

Julia M.I. Barth [1,12], Chrysoula Gubili [2,12], Michael Matschiner [3,4,12✉], Ole K. Tørresen [4], Shun Watanabe[5], Bernd Egger[1], Yu-San Han[6], Eric Feunteun[7,8], Ruben Sommaruga [9], Robert Jehle [10,13✉] & Robert Schabetsberger [11,13✉]

Genomic evidence is increasingly underpinning that hybridization between taxa is commonplace, challenging our views on the mechanisms that maintain their boundaries. Here, we focus on seven catadromous eel species (genus *Anguilla*) and use genome-wide sequence data from more than 450 individuals sampled across the tropical Indo-Pacific, morphological information, and three newly assembled draft genomes to compare contemporary patterns of hybridization with signatures of past introgression across a time-calibrated phylogeny. We show that the seven species have remained distinct for up to 10 million years and find that the current frequencies of hybridization across species pairs contrast with genomic signatures of past introgression. Based on near-complete asymmetry in the directionality of hybridization and decreasing frequencies of later-generation hybrids, we suggest cytonuclear incompatibilities, hybrid breakdown, and purifying selection as mechanisms that can support species cohesion even when hybridization has been pervasive throughout the evolutionary history of clades.

[1] Department of Environmental Sciences, Zoological Institute, University of Basel, Vesalgasse 1, 4051 Basel, Switzerland. [2] Fisheries Research Institute, Hellenic Agricultural Organisation-DEMETER, Nea Peramos, 64 007 Kavala, Greece. [3] Department of Palaeontology and Museum, University of Zurich, Karl-Schmid-Strasse 4, 8006 Zurich, Switzerland. [4] Centre for Ecological and Evolutionary Synthesis, Department of Biosciences, University of Oslo, P.O. Box 1066 Blindern0316 Oslo, Norway. [5] Faculty of Agriculture, Kindai University, 3327-204 Nakamachi, Nara 631-8505, Japan. [6] Institute of Fisheries Science, College of Life Science, National Taiwan University, No. 1, Sec. 4, Roosevelt Road, Taipei 10617, Taiwan. [7] Laboratoire Biologie des Organismes et Écosystèmes Aquatiques (BOREA), Muséum National d'Histoire Naturelle, CNRS, Sorbonne Université, Université de Caen Normandie, Université des Antilles, IRD, 61 Rue Buffon, CP 53, 75231 Paris Cedex 05, France. [8] MNHN—Station Marine de Dinard, Centre de Recherche et d'Enseignement Sur les Systèmes Côtiers (CRESCO), 38 Rue du Port Blanc, 35800 Dinard, France. [9] Department of Ecology, University of Innsbruck, Technikerstr. 25, 6020 Innsbruck, Austria. [10] School of Science, Engineering and Environment, University of Salford, Salford Crescent, Salford M5 4WT, UK. [11] Department of Biosciences, University of Salzburg, Hellbrunnerstrasse 34, 5020 Salzburg, Austria. [12] These authors contributed equally: Julia M. I. Barth, Chrysoula Gubili, Michael Matschiner. [13] These authors jointly supervised this work: Robert Jehle, Robert Schabetsberger. ✉email: michaelmatschiner@mac.com; R.Jehle@salford.ac.uk; robert.schabetsberger@sbg.ac.at

The turn of the century has witnessed a paradigm shift in how we view the role of hybridization for building up biological diversity. While hybridization was previously assumed to be spatially restricted and confined to a small number of taxa, it became gradually recognized that incomplete isolation of genomes is widespread across eukaryotes, with varied effects on adaptation and speciation[1–4]. More recently, this view has been further fuelled by technical and analytical advances that enable the quantification of past introgression—that is, the genetic exchange through hybridization—across entire clades, revealing that it is often the most rapidly diversifying clades that experienced high frequencies of introgression[5–8]. This seemingly paradoxical association between introgression and rapid species proliferation underlies a key question in evolutionary biology: How can species in diversifying clades be accessible for introgression but nevertheless solidify their species boundaries[9]? To answer this question, insights are required into the mechanisms that gradually reduce the degree to which hybridization generates introgression; however, these mechanisms are still poorly understood because contemporary hybridization and past introgression have rarely been studied and compared jointly across multiple pairs of animal species with different divergence times within a single clade[10].

Teleost fish provide well-established model systems to reveal processes of diversification, including the impact of hybridization on speciation[11,12]. A particularly promising system for hybridization research are catadromous freshwater eels of the genus Anguilla, one of the most species-rich genera of eels with high economic value[13]. These fishes are renowned for their unique population biology, whereby individuals of a given species migrate to one or only few oceanic spawning areas and reproduce panmictically within these[14–16]. Moreover, spawning is temporally and spatially overlapping between multiple species, which therefore are expected to have great potential for interspecies mating[17,18]. Frequent occurrence of hybridization has in fact been demonstrated with genomic data for the two Atlantic Anguilla species (A. anguilla and A. rostrata), with a particularly high proportion of hybrids in Iceland[15,19–22]. However, while these Atlantic species have so far received most of the scientific attention, the greatest concentration of Anguilla species is present in the tropical Indo-Pacific, where 11 species occur and may partially spawn at the same locations[23,24]. A locally high frequency of hybrids between two species occurring in this region (A. marmorata and A. megastoma) has been suggested by microsatellite markers and small datasets of species-diagnostic single-nucleotide polymorphisms (SNPs)[18]; however, the pervasiveness of hybridization across all tropical eel species, the degree to which hybridization leads to introgression in these species, and the mechanisms maintaining species boundaries have so far remained poorly known.

In the present paper, we use high-throughput sequencing and morphological analyses for seven species of tropical eels sampled across the Indo-Pacific to (i) infer their age and diversification history, (ii) determine the frequencies of contemporary hybridization between the species, (iii) quantify signatures of past introgression among them, and (iv) identify mechanisms that may be responsible for the stabilization of species boundaries. Our unique combination of approaches allows us to compare hybridization and introgression across multiple pairs of species with different ages and suggests that cytonuclear incompatibilities, hybrid breakdown, and purifying selection can strengthen species boundaries in the face of frequent hybridization.

## Results

**Extensive sampling.** Collected in 14 field expeditions over the course of 17 years, our dataset included 456 individuals from 14 localities covering the distribution of anguillid eels in the tropical Indo-Pacific (Fig. 1a, Supplementary Table 1). Whenever possible, eels were tentatively identified morphologically in the field. Restriction-site-associated DNA (RAD) sequencing for all 456 individuals resulted in a comprehensive dataset of 704,480 RAD loci with a mean of 253.4 bp per locus and up to 1,518,299 SNPs, depending on quality-filtering options (Supplementary Fig. 1). RAD sequencing reads mapping to the mitochondrial genome unambiguously assigned all individuals to one of the seven tropical eel species A. marmorata, A. megastoma, A. obscura, A. luzonensis, A. bicolor, A. interioris, and A. mossambica, in agreement with our morphological assessment that indicated that the remaining four Indo-Pacific Anguilla species A. celebesensis, A. bengalensis, A. borneensis, and A. reinhardtii were not included in our dataset (Supplementary Fig. 2). For those individuals for which sufficient morphological information was available (n = 161, restricted to A. marmorata, A. megastoma, A. obscura, and A. interioris), predorsal length without head length (PDH) and distance between the origin of the dorsal fin and the anus (AD), size-standardized by total length (TL)[25], showed species-specific clusters, even though these were not fully separated from each other (Fig. 1b, Supplementary Fig. 3). This diagnosis was further supported by principal component analysis (PCA) of seven morphological characters (Supplementary Fig. 3). After excluding individuals with low-quality sequence data, the sample set used for genomic analyses contained 430 individuals of the seven species, including 325 A. marmorata, 41 A. megastoma, 36 A. obscura, 20 A. luzonensis, 4 A. bicolor, 3 A. interioris, and 1 A. mossambica (Supplementary Tables 2, 3). The large number of individuals available for A. marmorata, A. megastoma, and A. obscura, sampled at multiple sites throughout their geographic distribution (Fig. 1a, Supplementary Table 1), permitted detailed analyses of genomic variation within these species (Supplementary Note 1). These analyses distinguished four populations in the geographically widespread species A. marmorata[26–29], but detected no population structure in A. megastoma and A. obscura (Supplementary Fig. 4), which are both presumed to have a single spawning area in the western South Pacific[18,30].

**Deep divergences among tropical eel species.** To analyze genomic variation among tropical eel species, we used a dataset of 155,896 SNPs derived from RAD sequencing (Supplementary Fig. 1) for PCA (Fig. 1c, Supplementary Fig. 5). With few exceptions (see next section), the 430 individuals grouped according to species, and the seven species included in our dataset formed largely well-separated clusters. Pairwise nuclear genetic distances between species ranged from 0.0053 to 0.0116 (uncorrected p-distance; excluding individuals with intermediate genotypes) and were largest for A. mossambica, followed by A. megastoma (Supplementary Table 4). We further investigated the relationships among tropical eels species and their divergence times by applying Bayesian phylogenetic inference to genome-wide SNPs[31], using the multispecies coalescent model implemented in the software SNAPP[32]. As SNAPP does not account for rate variation among substitution types, we performed separate analyses with transitions and transversions, both of which supported the same species-tree topology. In agreement with the pairwise genetic distances, A. mossambica appeared as the sister to a clade formed by all other species, and A. megastoma was resolved within this clade as the sister to a group formed by the species pair A. bicolor and A. obscura and the species trio A. marmorata, A. luzonensis, and A. interioris, with A. marmorata and A. luzonensis being most closely related within this trio (Fig. 1d, Supplementary Fig. 6). Each node of this species tree received full Bayesian support (Bayesian posterior probability, BPP, 1.0) and, except for the interrelationships of A. marmorata,

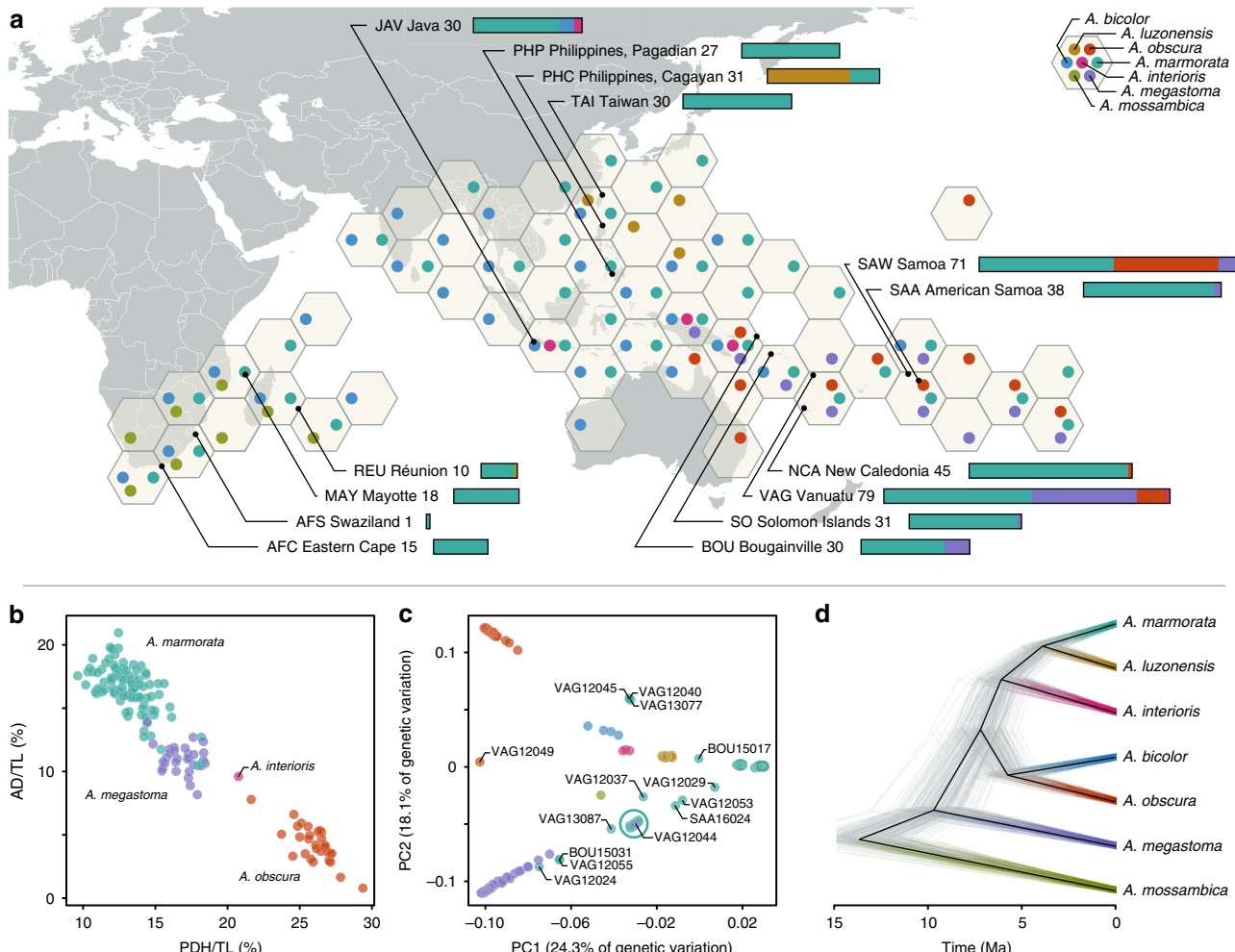

**Fig. 1 Genomic and morphological variation in tropical eels. a** Distribution of *Anguilla* species in the Indo-Pacific. The color and position of dots within hexagons indicate species presence within the region covered by the hexagon, according to the Global Biodiversity Information Facility database[122] and our own collection. Sampling locations are indicated with black dots. Numbers following location names specify the number of samples taken. Stacked bars indicate the species identities of individuals, according to mitochondrial and morphological species assignment. **b** Morphological variation among the four species *A. marmorata* ($n = 100$), *A. megastoma* ($n = 30$), *A. obscura* ($n = 30$), and *A. interioris* ($n = 1$). Dots represent individuals and are colored according to mitochondrial species identity. **c** Genomic principal component analysis (PCA) based on 155,896 variable sites. Specimen IDs are given for individuals with intermediate genotypes. The cyan circle indicates a cluster of 11 individuals mitochondrially assigned to *A. marmorata* (SAA16011, SAA16012, SAA16013, SAA16027, SAW17B27, SAW17B49, VAG12012, VAG12018, VAG12019, VAG13071, VAG13078), in addition to the highlighted VAG12044 that is mitochondrially assigned to *A. megastoma*. **d** Time-calibrated phylogeny based on 5000 transition sites. Each individual tree shown in gray represents a sample from the posterior tree distribution; a maximum-clade-credibility summary tree is shown in black. Color code in **b**, **c**, and **d** is identical to **a**. PC, principal component; AD, distance between the dorsal fin and the anus; PDH, predorsal length without head length; TL, total length.

*A. luzonensis*, and *A. interioris*, the tree agreed with previous phylogenies of mitochondrial sequences[33–37]. Using a published age estimate for the divergence of *A. mossambica*[14] to time calibrate the species tree, our analysis of transition SNPs with SNAPP showed that the clade combining all species except *A. mossambica* began to diverge around 9.7 Ma (divergence of *A. megastoma*; 95% highest-posterior-density interval, HPD, 11.7–7.7 Ma). This age estimate was robust to the use of transversions instead of transitions, alternative topologies enforced through constraints, and subsampling of taxa (Supplementary Fig. 6).

To allow for the integration into other timelines of eel diversification based on multimarker data[38,39], we used whole-genome sequencing (WGS) data and generated new draft genome assemblies for *A. marmorata*, *A. megastoma*, and *A. obscura* (N50 between 54,849 bp and 64,770 bp; Supplementary Table 5), and extracted orthologs of the markers used in the studies of Musilova

et al.[39] and Rabosky et al.[38]. The use of these combined datasets together with age calibrations from the two studies also had little effect on age estimates (Supplementary Fig. 6). Thus, all our analyses of divergence times point to an age of the clade formed by *A. marmorata*, *A. megastoma*, *A. obscura*, *A. luzonensis*, *A. bicolor*, and *A. interioris* roughly on the order of 10 Ma.

**High frequency of contemporary hybridization.** Despite their divergence times up to around 10 Ma, our genomic dataset revealed ongoing hybridization in multiple pairs of tropical eel species. Analyses of genomic variation with PCA revealed a number of individuals with genotypes intermediate to the main clusters formed by the seven species (Figs. 1c, 2a–d, Supplementary Fig. 5). The same individuals also appeared admixed in maximum-likelihood ancestry inference with the software

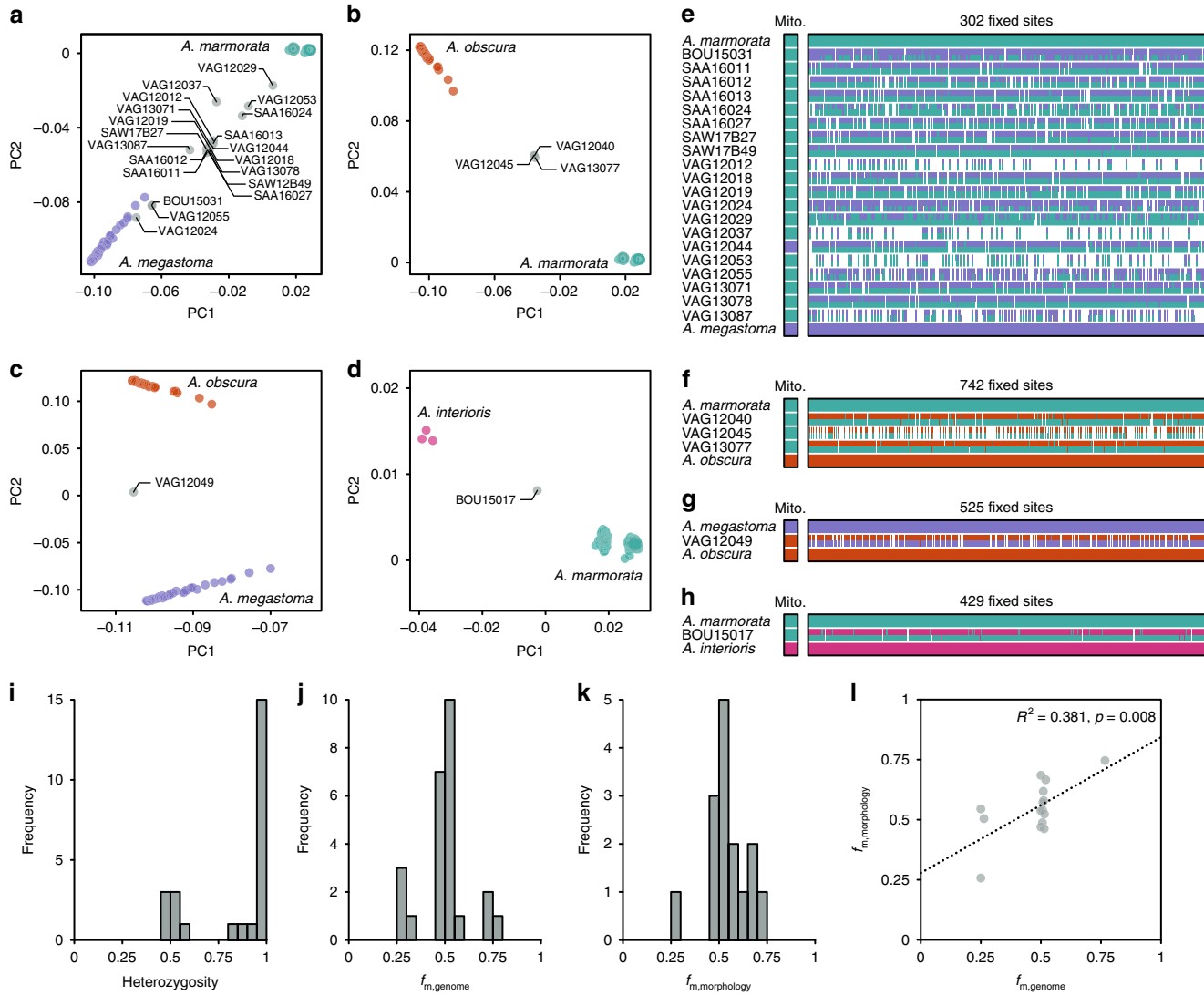

**Fig. 2 Contemporary hybridization among tropical eels. a–d** Genomic variation inferred in a PCA of all individuals of *A. marmorata*, *A. megastoma*, *A. obscura*, and *A. interioris* (Supplementary Fig. 5c), shown separately for four hybridizing species pairs: *A. marmorata* and *A. megastoma* (**a**), *A. marmorata* and *A. obscura* (**b**), *A. megastoma* and *A. obscura* (**c**), and *A. marmorata* and *A. interioris* (**d**). Individuals with intermediate positions are marked in gray with specimen IDs; species color code is as in Fig. 1. **e** Ancestry painting for 20 hybrids between *A. marmorata* and *A. megastoma*. The top and bottom horizontal bars represent 302 sites that are fixed for different alleles between the two species; all other bars indicate the alleles at each of those sites. White color indicates missing data. Heterozygous alleles are shown with the top half in each bar matching the second parental species and vice versa. **f** Ancestry painting for three contemporary hybrids between *A. marmorata* and *A. obscura*, based on 742 sites fixed between these two species. **g** Ancestry painting for one hybrid between *A. megastoma* and *A. obscura*, based on 525 fixed sites. **h** Ancestry painting for one hybrid between *A. marmorata* and *A. interioris*, based on 429 fixed sites. **i** Histogram of heterozygosity observed in hybrids. **j** Histogram of the proportions of hybrid genomes derived from the maternal species (according to mitochondrial sequence data). **k** Histogram of the relative morphological similarities between hybrids and the maternal species, measured as the relative proximity to the mean maternal phenotypes, compared to the proximity to the mean paternal phenotype. **l** Comparison of the proportions of hybrids' genomes derived from the maternal species and the similarity to the mean maternal species' phenotype. The dotted line indicates a significant positive correlation between the two measures (two-tailed *t* test; $t = 3.1$, $p = 0.008$, $R^2 = 0.381$). PC, principal component; mito., mitochondrial genome; AD, distance between the dorsal fin and the anus; PDH, predorsal length without head; TL, total length.

ADMIXTURE[40] (Supplementary Fig. 7, Supplementary Table 6) and had high levels of coancestry with two other species in analyses of RAD haplotype similarity with the program fineRADstructure, indicative of hybrid origin[41] (Supplementary Fig. 8). In contrast to these signals of interspecific hybridization, no *A. marmorata* individuals had genotypes clearly intermediate between the four distinct *A. marmorata* populations (Supplementary Fig. 4).

For each of the putative interspecific hybrid individuals, we produced ancestry paintings[42] based on sites that are fixed for different alleles in the parental species. In these ancestry paintings, the genotypes of the putative hybrids are assessed for those sites fixed between parents, with the expectation that first-generation (F1) hybrids should be heterozygous at almost all of these sites, and backcrossed hybrids of the second generation should be heterozygous at about half of them. All of the putative hybrids were confirmed by the ancestry paintings, showing that our dataset includes 20 hybrids between *A. marmorata* and *A. megastoma*, 3 hybrids between *A. marmorata* and *A. obscura*, 1 hybrid between *A. megastoma* and *A. obscura*, and 1 hybrid between *A. marmorata* and *A. interioris* (Fig. 2e–h, Supplementary Figs. 9–13, Supplementary Table 7). The frequency of

hybrids in our dataset is thus 5.8% overall and up to 22.5% at the hybridization hotspot of Gaua, Vanuatu[18] (Supplementary Fig. 14, Supplementary Table 8). This high frequency is remarkable, given that most animal species produce hybrids at a frequency far below 1%[1,43]. The heterozygosities of the hybrids are clearly bimodal with a peak near 1 and another around 0.5 (Fig. 2i, Supplementary Table 7), supporting the presence of both first-generation hybrids and backcrossed second-generation hybrids.

Using the mitochondrial genomes of hybrids as an indicator of their maternal species, we quantified the proportions of their nuclear genomes derived from the maternal species, $f_{m,genome}$, based on their genotypes at the fixed sites used for ancestry painting. The distribution of these $f_{m,genome}$ values has three peaks centered around 0.25 (4 individuals), 0.5 (18 individuals), and 0.75 (3 individuals), suggesting that backcrossing has occurred about equally often with both parental species (Fig. 2j, Supplementary Fig. 15).

In their size-standardized overall morphology, all hybrids for which morphological information was available ($n = 15$) were intermediate between the two parental species (Supplementary Fig. 16). Following Watanabe et al.[25], we measured this overall morphology by the ratios AD/TL and PDH/TL, where AD is the distance between the dorsal fin and the anus, TL is the total length, and PDH is the predorsal length without the head. From these two ratios, we quantified the morphological similarity of hybrids to their maternal species relative to their paternal species, $f_{m,morphology}$, as their position on an axis connecting the mean phenotypes of the two parental species (Supplementary Fig. 16). Similar to the distribution of $f_{m,genome}$ values (Fig. 2j), the distribution of $f_{m,morphology}$ values (Fig. 2k) also has three peaks centered close to 0.25, 0.5, and 0.75, and the two values were correlated (Fig. 2l). In contrast to their intermediate size-standardized overall morphology, hybrids in some cases had certain transgressive characters, exceeding the range of the parental phenotypes[44,45] (Supplementary Figs. 17, 18). This was the case for VAG13071 and VAG12044, two F1 hybrids between A. marmorata and A. megastoma that had the greatest total length among all sampled individuals (Supplementary Table 1, Supplementary Fig. 17).

Notably, we recorded no signals of hybridization with Anguilla species that were not included in our dataset. Anguilla celebesensis, A. bengalensis, A. borneensis, and A. reinhardtii all occur in the Indo-Pacific and could be expected to hybridize with the sampled species. If our dataset had included hybrids between sampled and unsampled species, we could have identified these hybrids from their expected increased heterozygosity, outlier positions in PCA analyses, and separate clustering in our analyses with ADMIX-TURE and fineRADstructure. However, as hybridization with the unsampled species could be locally restricted away from the sampled localities, a more extensive sampling scheme might be required to assess its overall frequency.

**Signatures of past introgression.** Multiple independent approaches revealed highly variable signatures of past introgression among species pairs of tropical eels. First, we found discordance between the Bayesian species tree based on the multispecies coalescent model (Fig. 1d) and an additional maximum-likelihood tree inferred with IQ-TREE[46] (Supplementary Fig. 19) from 1360 concatenated RAD loci selected for high SNP density (Supplementary Fig. 1). Even though both types of trees received full node support, their topologies differed in the position of A. interioris, which appeared next to A. marmorata and A. luzonensis in the Bayesian species tree (Fig. 1d, Supplementary Fig. 6), but as the sister to A. bicolor and A. obscura in the maximum-likelihood tree, in agreement with mitochondrial

phylogenies[14,34]. We applied an approach recently implemented in IQ-TREE[47] to assess per-locus and per-site concordance factors as additional measures of node support in the maximum-likelihood tree. These concordance factors were substantially lower than bootstrap-support values and showed that as few as 4.7% of the individual RAD loci and no more than 39.7% of all sites supported the position of A. interioris as the sister to A. bicolor and A. obscura.

To further test whether the tree discordance is due to past introgression or other forms of model misspecification, we applied genealogy interrogation[48], comparing the likelihood of different topological hypotheses for each of the 1360 RAD loci (Fig. 3a). We find that neither the topology of the Bayesian species trees nor the topology of the maximum-likelihood tree received most support from genealogy interrogation. Instead, 773 loci (62% of the informative loci) had a better likelihood when A. interioris was the sister to a clade formed by A. marmorata, A. luzonensis, A. bicolor, and A. obscura, compared to the topology of the Bayesian species tree (A. interioris as the sister to A. marmorata and A. luzonensis; Fig. 1d). The position of A. interioris as the sister to the other four species also had a better likelihood than the topology of the maximum-likelihood tree (A. interioris as the sister to A. bicolor and A. obscura; Supplementary Fig. 19) for 659 loci (53% of the informative loci). We thus assumed that the topology supported by genealogy interrogation (with A. interioris being the sister to A. marmorata, A. luzonensis, A. bicolor, and A. obscura) is our best estimate of the true species-tree topology. However, we observed an imbalance in the numbers of loci supporting the two alternative topologies, as 541 loci had a better likelihood when A. interioris was the sister to A. marmorata and A. luzonensis, whereas 685 loci had a better likelihood when A. interioris was the sister to A. bicolor and A. obscura (Fig. 3a). As incomplete lineage sorting would be expected to produce equal support for both alternative topologies but the imbalance is too large to arise stochastically (two-tailed binomial test; $p < 10^{-4}$), genealogy interrogation supports past introgression among A. interioris, A. bicolor, and A. obscura.

We further quantified both Patterson's D statistic[49,50] and the $f_4$ statistic[51,52] from biallelic SNPs, for all species quartets compatible with the species tree supported by genealogy interrogation. Both of these statistics support past introgression when they are found to differ from zero. We found that the $f_4$ statistic was significant in no less than 29 out of 60 species quartets (Supplementary Table 9). The most extreme D and $f_4$ values were observed in quartets in which A. mossambica was in the outgroup position, A. marmorata was in the position of the unadmixed species (P1), and A. interioris was in a position (P3) sharing gene flow with either A. luzonensis ($D = 0.41$) or A. bicolor ($f_4 = -0.011$) (P2). The sum of the analyses of D and $f_4$ suggests pervasive introgression among tropical eel species (Table 1), with significant support for gene flow between A. interioris and each of the four species A. luzonensis, A. bicolor, A. obscura, and A. megastoma, between A. luzonensis and both A. bicolor and A. obscura, and between A. marmorata and A. bicolor (Fig. 3b). While the pervasiveness of these signals prevents a clear resolution of introgression scenarios, the patterns could potentially be explained by a minimum of five introgression events: introgression between A. megastoma and A. interioris, between A. interioris and the common ancestor of A. bicolor and A. obscura, between A. interioris and A. luzonensis, between A. luzonensis and the common ancestor of A. bicolor and A. obscura, and between A. bicolor and A. marmorata (Fig. 3b). The four different populations of A. marmorata all showed nearly the same signal of gene flow with A. bicolor, indicating that either the introgression between these species predates the origin of the

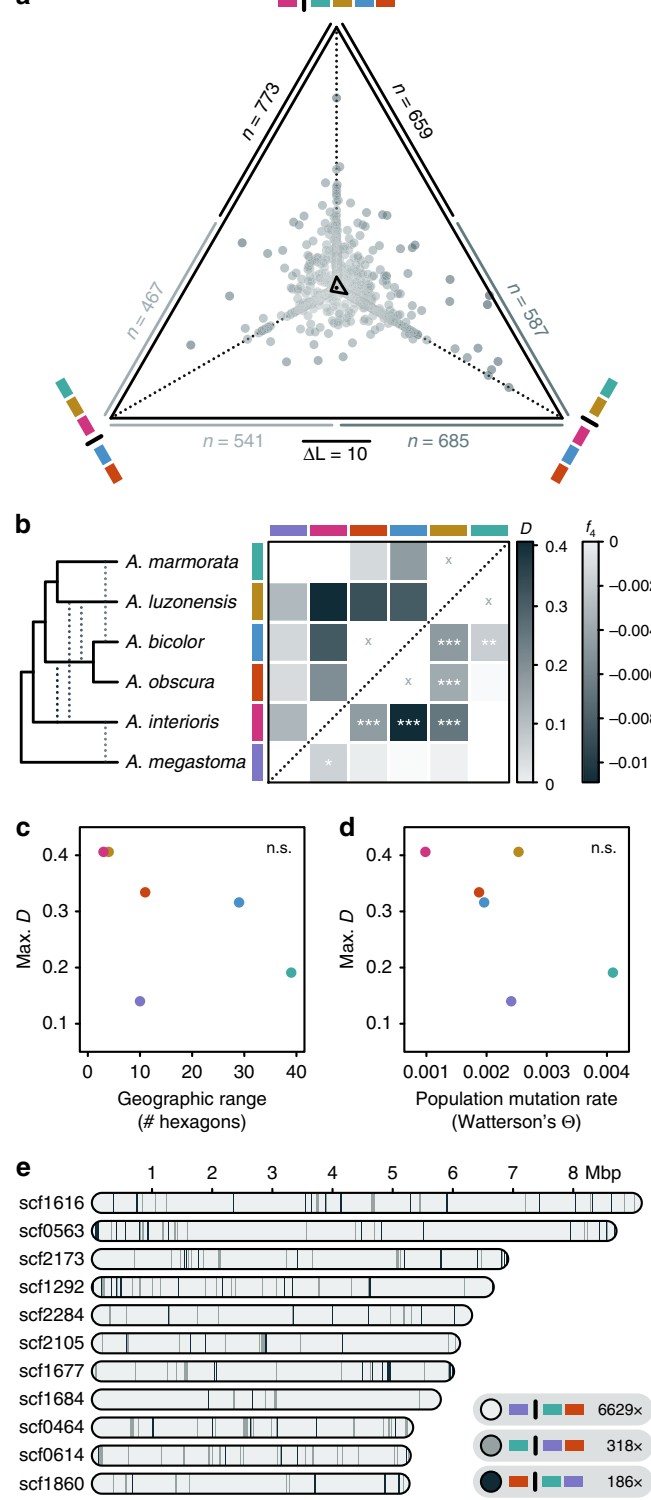

**Fig. 3 Past introgression among tropical eels. a** Likelihood support of individual RAD loci for different relationships of *A. interioris*: as sister to *A. marmorata* and *A. luzonensis* (bottom left), as sister to *A. obscura* and *A. bicolor* (bottom right), and as sister to a clade formed by those four species (top). The position of each dot shows the relative likelihood support of one RAD locus for each of the three tested relationships, with a distance corresponding to a log-likelihood difference of 10 indicated by the scale bar. The central triangle connects the mean relative likelihood support for each relationship. A black dot inside that triangle marks the central position corresponding to equal support for all three relationships. Sample sizes (*n*) report the number of loci that support each of the two competing relationships connected by that edge. **b** Heatmap indicating maximum pairwise *D* (above diagonal) and $f_4$ (below diagonal) statistics (see Table 1). Combinations marked with "x" symbols indicate sister taxa; introgression between these could not be assessed. Asterisks indicate the significance of $f_4$ values (*$p < 0.05$; **$p < 0.01$; ***$p < 0.001$; not adjusted for multiple comparisons; see Table 1 for precise values), determined through one-sided comparison with coalescent simulations with the F4 software[52]. The cladogram on the left summarizes the species-tree topology according to **a** and the significant signals of introgression according to **b**. **c**, **d** Comparisons of the maximum *D* value per species with the species' geographic range or population mutation rate Θ. Geographic range was measured as the number of geographic hexagons (see Fig. 1) in which the species is present, and Watterson's estimator[123] was used for the population mutation rate Θ. n.s. not significant. **e** Genomic patterns of phylogenetic relationships among *A. marmorata*, *A. obscura*, and *A. megastoma*, based on WGS reads mapped to the 11 largest scaffolds (those longer than 5 Mbp) of the *A. anguilla* reference genome. Blocks in light gray show 20,000-bp regions (incremented by 10,000 bp) in which *A. marmorata* and *A. obscura* appear as sister species, in agreement with the inferred species tree; in other blocks, *A. megastoma* appears closer to either *A. obscura* (gray) or *A. marmorata* (dark gray).

effective population size $N_e$ (as $\Theta = 4N_e\mu$; with $\mu$ being the mutation rate), are those with the weakest signals of introgression (Fig. 3d) despite a high frequency of hybrids between them. This observation could be explained if introgressed alleles are over time more effectively purged by purifying selection from the genomes of species with larger effective population sizes[9,53–55]. Particularly large effective population sizes in *A. marmorata* and *A. megastoma* are in fact supported by the WGS data produced for one individual of both species as well as *A. obscura*. When analyzed with the pairwise sequentially Markovian coalescent model[56], these data yielded estimates of a contemporary $N_e$ between $9.9 \times 10^4$ and $6.0 \times 10^5$ for *A. marmorata* and between $2.3 \times 10^5$ and $2.0 \times 10^6$ for *A. megastoma*, whereas a comparatively lower $N_e$ between $3.4 \times 10^4$ and $7.4 \times 10^4$ was estimated for the third species with WGS data, *A. obscura* (Supplementary Fig. 20).

Low levels of introgression in the genomes of *A. marmorata* and *A. megastoma* were also supported by these WGS data. Aligning the WGS reads of *A. marmorata*, *A. megastoma*, and *A. obscura* to the *A. anguilla* reference genome assembly[57] resulted in an alignment with 23,165,451 genome-wide SNPs. Based on these SNPs, and using *A. anguilla* as the outgroup, the *D* value supporting gene flow between *A. marmorata* and *A. megastoma* was only 0.007 (Table 1). Phylogenetic analyses for 7133 blocks of 20,000 bp, incremented by 10,000 bp, on the 11 largest scaffolds of the *A. anguilla* assembly showed that as many as 6629 blocks (93%) support the species-tree topology, in which *A. marmorata* and *A. obscura* appear more closely related to each other than to *A. megastoma* (Fig. 3e). The alternative topologies with either *A. obscura* or *A. marmorata* being closer to *A. megastoma* were supported by 318 (4%) and 186 (3%) blocks,

observed spatial within-species differentiation in *A. marmorata*, or that each *A. marmorata* population had gene flow with the similarly widespread *A. bicolor* (Supplementary Table 10).

Interestingly, it appears that the species with the most restricted geographic distributions—*A. interioris* and *A. luzonensis*—are those with the strongest signals of past introgression (Fig. 3c), even though we identified only a single instance of contemporary hybridization involving one of these species (Fig. 2d, h). In contrast, *A. marmorata* and *A. megastoma*, which both have a high population mutation rate Θ indicative of a large

**Table 1 Past introgression supported by $D$ and $f_4$ statistics.**

| P1 | P2 | P3 | $n$ | $C_{ABBA}$ | $C_{BABA}$ | $D$ | $f_4$ | $p$ |
|----|----|----|----|----|----|----|----|----|
| A. marmorata | A. luzonensis | A. interioris | 10,290 | 182.7 | 77.1 | 0.406 | −0.0070 | 0.000 |
| A. marmorata | A. luzonensis | A. obscura | 15,689 | 186.6 | 93.0 | 0.334 | −0.0043 | 0.000 |
| A. marmorata | A. bicolor | A. interioris | 7772 | 266.3 | 138.4 | 0.316 | −0.0109 | 0.000 |
| A. marmorata | A. luzonensis | A. bicolor | 11,542 | 158.1 | 82.8 | 0.313 | −0.0052 | 0.000 |
| A. marmorata | A. obscura | A. interioris | 10,208 | 307.9 | 197.8 | 0.218 | −0.0051 | 0.000 |
| A. obscura | A. bicolor | A. interioris | 8304 | 123.8 | 84.1 | 0.191 | −0.0030 | 0.005 |
| A. obscura | A. bicolor | A. marmorata | 11,372 | 104.7 | 71.2 | 0.191 | −0.0025 | 0.002 |
| A. obscura | A. bicolor | A. luzonensis | 12,557 | 113.4 | 80.0 | 0.173 | −0.0022 | 0.008 |
| A. marmorata | A. interioris | A. megastoma | 9951 | 96.4 | 72.7 | 0.140 | −0.0023 | 0.026 |
| A. marmorata | A. luzonensis | A. megastoma | 13,129 | 69.0 | 52.9 | 0.133 | −0.0008 | 0.201 |
| A. luzonensis | A. marmorata | A. bicolor | 14,675 | 105.4 | 84.5 | 0.110 | −0.0011 | 0.106 |
| A. luzonensis | A. bicolor | A. interioris | 14,246 | 228.4 | 191.0 | 0.089 | −0.0015 | 0.062 |
| A. luzonensis | A. interioris | A. megastoma | 13,632 | 82.4 | 70.2 | 0.080 | −0.0007 | 0.192 |
| A. marmorata | A. bicolor | A. megastoma | 11,134 | 110.9 | 95.0 | 0.077 | −0.0003 | 0.430 |
| A. luzonensis | A. marmorata | A. obscura | 15,500 | 111.7 | 96.5 | 0.073 | −0.0003 | 0.406 |
| A. marmorata | A. obscura | A. megastoma | 11,647 | 126.1 | 110.0 | 0.068 | −0.0009 | 0.241 |
| A. bicolor | A. obscura | A. marmorata | 11,303 | 80.0 | 73.0 | 0.046 | −0.0007 | 0.261 |
| A. obscura | A. bicolor | A. megastoma | 11,761 | 64.7 | 59.5 | 0.042 | −0.0002 | 0.447 |
| A. bicolor | A. obscura | A. luzonensis | 15,856 | 78.1 | 72.1 | 0.040 | −0.0010 | 0.141 |
| A. obscura | A. interioris | A. megastoma | 11,017 | 96.2 | 90.8 | 0.029 | −0.0011 | 0.137 |
| A. luzonensis | A. bicolor | A. megastoma | 14,602 | 97.0 | 93.1 | 0.020 | 0.0002 | 0.416 |
| A. bicolor | A. interioris | A. megastoma | 10,451 | 84.0 | 82.0 | 0.012 | −0.0007 | 0.213 |
| A. luzonensis | A. obscura | A. interioris | 15,143 | 227.1 | 221.7 | 0.012 | 0.0005 | 0.300 |
| A. luzonensis | A. obscura | A. megastoma | 15,405 | 107.9 | 106.2 | 0.008 | −0.0001 | 0.461 |
| A. obscura | A. marmorata | A. megastoma | 23,165,451 | 596,786.0 | 587,910.0 | 0.007 | — | — |

Only comparisons that are compatible with the inferred phylogenetic relationships and result in positive $D$ values are shown (for all comparisons, see Supplementary Table 9). All except the comparison in the last row are based on RAD-sequencing-derived SNP data; the last comparison is based on WGS reads of a single individual of the three species. Either *A. mossambica*, *A. megastoma*, *A. interioris*, or *A. anguilla* (in the comparison based on WGS data) were used as outgroups and the comparison resulting in the largest $D$ value is reported when multiple of these outgroups were used. $p$ values are based on one-sided comparisons and not adjusted for multiple comparisons. $n$: number of sites variable among the included species; $C_{ABBA}$: number of sites at which species P2 and P3 share the derived allele; $C_{BABA}$: number of sites at which P1 and P3 share the derived allele. Italic font is used for species names and variables.

respectively. Notably, we did not observe long sets of adjacent blocks supporting the alternative topologies, which would be expected if the individuals had hybrids in their recent ancestry[58]. The longest set of blocks supporting *A. marmorata* and *A. megastoma* as most closely related encompassed merely 80,000 bp (positions 4,890,000 to 4,970,000 on scaffold scf1677). While the lack of phasing information and a recombination map prevents a statistical test of time since admixture[58], the absence of longer sets of blocks most likely excludes hybrid ancestors within the last 10–20 generations.

**Evidence of cytonuclear incompatibility**. With a single exception, all of the 20 hybrids between *A. marmorata* and *A. megastoma* possessed the mitochondrial genome of *A. marmorata*, indicating that it is almost exclusively female *A. marmorata* that are involved in successful hybridization events (Fig. 2e). None of the seven backcrosses possessed the *A. megastoma* mitochondrial genome, and thus the mother of the mother of each backcross must have been an *A. marmorata*. Such asymmetry indicates differential viability of hybrids depending on the directionality of mating and could result from cytonuclear incompatibilities[20,59–62].

To identify potential causes of cytonuclear incompatibility between the two species, we investigated their nuclear and mitochondrial genomes for rearrangements within genes and for nonsynonymous substitutions between the species. Pairwise whole-genome alignment of the *A. marmorata* and *A. megastoma* genome assemblies with the *A. anguilla* reference genome assembly revealed at least one clear example of a large-scale (>1 kb) inversion in *A. megastoma* and several further putative inversions and transpositions (Supplementary Table 11, Supplementary Fig. 21). Mapping of *A. megastoma* WGS reads to the genome assembly of the same species further indicated the heterozygous presence of an inversion with a length of about 8 kb. This inversion changes the orientation of at least two regions homologous to exons of the zebrafish (*Danio rerio*) *myhc4* gene (NCBI accession NM_001020485), suggesting that the inversion affects the protein encoded by this gene, myosin heavy chain, in part of the *A. megastoma* population (Fig. 4a, Supplementary Fig. 22, Supplementary Table 12).

A closer inspection of the RAD-sequencing-derived nuclear sites fixed between *A. marmorata* and *A. megastoma* (Fig. 2e) showed that nine of the 302 fixed sites lie within coding regions, according to gene prediction for the *A. anguilla* genome assembly with AUGUSTUS[63] (Supplementary Table 13). Of these nine fixed differences, three change an amino acid in the translation of the predicted gene, including, in one case, a change in a region homologous to exon 191 of the zebrafish *ttna* gene, encoding for titin (NCBI accession DQ649453) (Fig. 4b, Supplementary Table 14).

Finally, by comparing the mitochondrial genomes of *A. marmorata* and *A. megastoma*, we identified 67 mitochondrial amino-acid changes between the two species (Fig. 4c, Supplementary Table 15; whether or not these changes were fixed in the two species could not be determined as we only had mitochondrial genome information of one individual per species). The greatest density of these changes was found in the translation of the *mt-atp6* gene, where 15 out of 227 amino acids (6.6%) were different between the two species.

Based on these findings, we propose that differences in *myhc4*, *ttna*, and *mt-atp6* could be possible causes of cytonuclear incompatibility between *A. marmorata* and *A. megastoma*. The products of *myhc4* and *ttna*, myosin heavy chain and titin, are both essential for muscle function[64,65] and their joint work is powered by hydrolysis of adenosine triphosphate (ATP) in the myosin heavy chain subunit. In turn, ATP is produced at mitochondrial

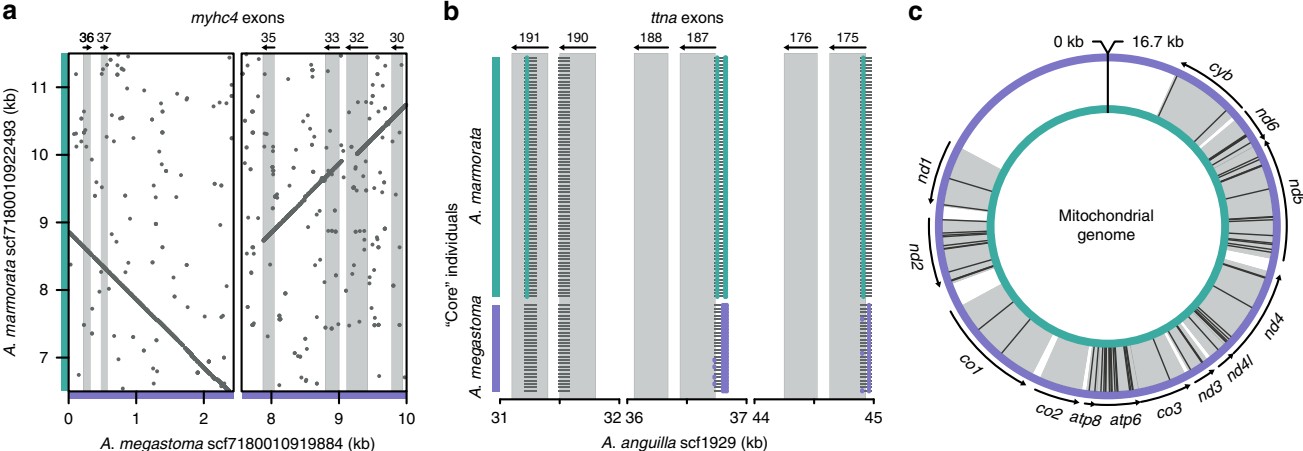

**Fig. 4 Potential causes of cytonuclear incompatibility between *A. marmorata* and *A. megastoma*. a** Dot plot visualizing an inversion between *A. megastoma* scaffold scf7180010919884 (purple) and the homologous *A. marmorata* scaffold scf7180010922493 (cyan). Each dot represents a tuple of nine nucleotides that are identical or the reverse complement between the two scaffolds. Light gray rectangles mark regions homologous to exons of the zebrafish (*Danio rerio*) *myhc4* gene; order and orientation of these exons are indicated at the top (Supplementary Table 12). The inversion may thus affect the transcript of this gene in *A. megastoma*. Other detected genomic rearrangements are shown in Supplementary Figs. 21 and 22. **b** Illustration of fixed sites in coding sequences between *A. marmorata* and *A. megastoma*. Light gray rectangles mark regions homologous to exons of the zebrafish *ttna* gene (Supplementary Table 14). Dark gray lines indicate RAD sequences for 72 and 26 "core" individuals of *A. marmorata* and *A. megastoma*, respectively. Nonreference genotypes are colored according to the species in which the genotype is most frequently found. Within exon 191, an adenine to guanosine substitution in *A. marmorata* at position 32,226 of *A. anguilla* scaffold scf1929 changes amino acid 5732 of the protein from aspartic acid to glycine. Other protein-coding sites fixed between *A. marmorata* and *A. megastoma* are listed in Supplementary Table 13. **c** Nonsynonymous substitutions between the *A. marmorata* (inner, cyan circle) and *A. megastoma* (outer, purple circle) mitochondrial genomes. Gray segments indicate protein-coding genes and nonsynonymous substitutions within these are shown with black lines. All mitochondrial amino-acid changes are listed in Supplementary Table 15.

membranes by the enzyme ATP synthase, which is in part encoded by *mt-atp6*[66]. It could therefore be possible that modifications in ATP synthase in one of the two species, influencing for example the efficiency of ATP production, are incompatible with altered properties of myosin and titin in the other species, which could reduce the fitness of hybrids in which the modifications co-occur. This type of fitness reduction could be particularly relevant in species that rely on highly efficient muscle function, such as anguillid eels during their oceanic spawning migrations. Interestingly, myosin heavy chain, titin, and ATP synthase have also been linked to cytonuclear incompatibility in the two Atlantic eel species *A. anguilla* and *A. rostrata*, where myosin heavy chain and titin were among 94 proteins with fixed sites between the two species and *mt-atp6* was found to be under strong positive selection together with a nuclear interactor gene[20,62].

## Discussion

As species diverge, genetic incompatibilities accumulate[67–69] and reduce the viability of hybrids[70]. However, the absolute timescale on which hybrid inviability evolves vastly exceeds the ages of species in many diversifying clades, indicating that species boundaries in these groups are maintained by reproductive barriers that act after the F1 stage[60,61,71–76]. For anguillid eels, laboratory experiments have produced hybrids between several species pairs, including *A. anguilla* and *A. australis*[77], *A. anguilla* and *A. japonica*[78,79], and *A. australis* and *A. dieffenbachii*[80]. These species pairs result from some of the earliest divergence events within the genus (Supplementary Fig. 6), suggesting that the limits of hybrid viability are not reached in anguillid eels. Our observation of frequent hybridization in four different species pairs, including two pairs involving *A. megastoma* with a divergence time around 10 Ma (Fig. 1d), supports this conclusion in a natural system, indicating that prezygotic reproductive barriers may generally be weak in tropical eels. This interpretation is strengthened by the fact that the 25 hybrids in our dataset were sampled in five

different years (Supplementary Table 7), suggesting that natural hybridization in tropical eels occurs continuously, rather than, for example, being the result of an environmental trigger that ephemerally caused spatially and temporally overlapping spawning[81]. Moreover, the seven identified backcrosses demonstrate that hybrids, at least those between *A. marmorata* and *A. megastoma*, can successfully reproduce naturally, indicating that, just like prezygotic barriers, postzygotic barriers (in the form of a reduction of F1 fertility) are also incomplete in tropical eels, even after 10 million years of divergence.

Nevertheless, by considering both hybridization frequencies and introgression signals across multiple species pairs, our analyses reveal how species collapse has been prevented in tropical eel species despite their great potential for genomic homogenization. First, asymmetry in the inheritance of mitochondrial genomes in hybrids suggests cytonuclear incompatibility between *A. marmorata* and *A. megastoma*, which is supported by our identification of genetic differences related to muscle function in the two species. Second, the lower frequency of backcrosses compared to F1 hybrids and the lack of later-generation backcrosses also suggest decreased fitness of hybrids. This hypothesis is supported by the observation that the *A. marmorata* and *A. megastoma* individuals selected for WGS apparently did not have recent hybrid ancestors, even though these individuals were sampled at the hybridization hotspot of Gaua, Vanuatu, where over 20% of all specimens are hybrids (Supplementary Table 8). Thus, it is possible that hybrid breakdown, affecting the viability and fertility of later-generation hybrids to a greater extent than F1 hybrids[74,76,82], is common in tropical eels and reduces the amount of introgression generated by backcrossing. Finally, the degree of introgression present in the genomes of tropical eel species appears to depend more on their population sizes than their hybridization frequencies, which could suggest that most introgressed alleles are purged from the recipient species by purifying selection[9,53–55]. This purging may be particularly

effective in tropical eels due to their largely panmictic populations, preventing deleterious alleles from persisting in isolated subgroups.

The combination of cytonuclear incompatibility, hybrid breakdown, and purifying selection may thus effectively reduce gene flow among tropical eels to a trickle that is too weak to break down species boundaries. Over the last 10 million years, this trickle might nevertheless have contributed to the diversification of tropical eels by providing the potential for adaptive introgression[3,83], which could for example have aided local adaptation following range expansion. Due to their unique catadromous life cycle, speciation in anguillid eels is assumed to proceed in one of two ways: either through a gradual expansion of the freshwater growth habitat, followed by reproductive isolation when spawning areas become separated in space or time, or through the transport of larvae into a different ocean current system—perhaps due to changes in palaeooceanographic conditions—followed by the establishment of a new spawning area in that system[22,26]. The latter process may be responsible for the differentiation of geographically separated populations or subspecies within *A. marmorata*, *A. bicolor*, and *A. bengalensis*[24]. Particularly during the early colonization stages in a new system, introgression from other species already established in that system may be beneficial for local adaptation, leaving their signatures in the genome. The identification of such signatures based on population-level whole-genome resequencing in tropical eels will be a promising goal for future studies.

## Methods

**Sample collection**. A total of 456 *Anguilla* specimens were obtained from 14 main localities over 17 years (2001−2017; Fig. 1, Supplementary Table 1). Sampling localities included South Africa (AFC: $n = 16$), Swaziland (AFS: $n = 1$), Mayotte (MAY: $n = 18$), Réunion (REU: $n = 10$), Indonesia (JAV: $n = 30$), Philippines (PHC/PHP: $n = 58$), Taiwan (TAI: $n = 30$), Bougainville Island (BOU: $n = 30$), Solomon Islands (SOK/SOL/SON/SOR/SOV: $n = 31$), Vanuatu (VAG: $n = 79$), New Caledonia (NCA: $n = 45$), Samoa (SAW: $n = 71$), and American Samoa (SAA: $n = 38$). Sampling was performed by electrofishing and with handnets in estuaries, rivers, and lakes, targeting elvers, yellow eels, and silver eels. Small fin clips were extracted from the pectoral fin of each specimen and stored in 98% ethanol, to be used in subsequent genetic analyses. Permits were obtained prior to sampling from the responsible authorities. The project was approved by the Research, Innovation and Academic Engagement Ethical Approval Panel of the University of Salford (the institution where DNA extraction and library preparation took place, permit number ST15/68). Local governments further approved the sampling protocols.

**Morphological analyses**. Morphological variation was assessed based on the following measurements: total length (TL), weight, preanal length (PA), predorsal length (PD), head length (HL), mouth length, eye distance, eye size (horizontal and vertical), pectoral fin size, head width, and girth[25]. We further calculated the distance between the anus and the dorsal fin (AD = PA − PD), predorsal length without head length (PDH = PD − HL), tail length (T = TL − PA), and preanal length without head length (TR = PA − HL). Morphological variation was assessed with PCA in the program JMP v.7.0 (SAS Institute Inc.; www.jmp.com) based on the ratios of PA, T, HL, TR, PD, PDH, and AD to TL; this analysis was performed for 161 individuals for which all measurements were available (100 *A. marmorata*, 30 *A. megastoma*, 30 *A. obscura*, and 1 *A. interioris*). Principal component scores were used to delimit "core" groups of putatively unadmixed individuals for the three species *A. marmorata* (73 individuals), *A. megastoma* (26 individuals), and *A. obscura* (26 individuals). In addition to PCA, we plotted the ratios of AD and PDH to TL, which were found to be particularly diagnostic for *Anguilla* species[84].

**Sequencing and quality filtering**. Genomic DNA was extracted using the DNeasy Blood and Tissue Kit (Qiagen) as per the manufacturer's instructions, or using a standard phenol chloroform procedure[85]. DNA quality of each sample was evaluated on an agarose gel and quantified on a Qubit Fluorometer 2.0 (Thermo Fisher Scientific). Double-digest restriction-site-associated DNA sequencing (ddRAD) was completed following Peterson et al.[86] with minor modifications; this protocol is described in Supplementary Note 2. Paired-end Illumina HiSeq 4000 sequencing was performed at Macrogen (Korea).

Returned demultiplexed reads were processed using the software Stacks v.2.0-beta9 and v.2.2[87], following the protocol described by Rochette and Catchen[88]. In brief, the reads were checked for correct cut sites and adaptor sequences using the "process_radtags" tool and subsequently mapped against the European eel

(*A. anguilla*) genome assembly[57] using BWA MEM v.0.7.17[89]. As this assembly does not include the mitochondrial genome, mitochondrial reads were identified by separately mapping against the *A. japonica* mitochondrial genome (NCBI accession CM002536). Mapped reads were sorted and indexed using SAMtools v.1.4[90,91]. Species identification was verified for all individuals by comparing mitochondrial sequences with the NCBI Genbank database using BLAST v.2.7.1[92]. Individuals with low-quality sequence data (with a number of reads below 600,000, a number of mapped reads below 70%, or a proportion of singletons above 5%) were excluded ($n = 26$). Variants were called using the "gstacks" tool, requiring a minimum mapping quality of 20 and an insert size below 500. Called variants were exported to variant call format (VCF) and haplotype format using the "populations" tool, allowing maximally 20% missing data and an observed heterozygosity below 75%, returning 1,518,299 SNPs.

The VCF file was further processed in two separate ways to generate suitable datasets for phylogenetic and population genetic analyses based on SNPs. For phylogenetic analyses, the VCF file was filtered with BCFtools v.1.6[91] to mask genotypes if the per-sample read depth was below 5 or above 50 or if the genotype quality was below 30. Sites were excluded from the dataset if they appeared no longer polymorphic after the above modifications, if genotypes were missing for 130 or more of the 460 individuals (30%), or if their heterozygosity was above 50%. The resulting VCF file contained 619,353 SNPs (Supplementary Fig. 1).

For analyses of genomic variation within and among genomic regions, filtering was done using VCFtools v.0.1.14[93] and PLINK v.1.9[94]. Sites were excluded if the mean read depth was above 50, the minor allele frequency was below 0.02, or heterozygosity excess was supported with $p < 0.05$ (rejecting the null hypothesis of no excess). In addition, individual genotypes were masked if they had a read depth below 5 or a genotype quality below 30. The resulting VCF file contained 155,896 SNPs (Supplementary Fig. 1).

For each of the three species *A. marmorata*, *A. megastoma*, and *A. obscura*, one individual (VAG12030, VAG12032, and VAG12050, respectively) sampled in Gaua, Vanuatu, was subjected to WGS. Genomic DNA was extracted using the DNeasy Blood and Tissue Kit (Qiagen) according to the manufacturer's protocol. DNA quality was evaluated on an agarose gel and quantified on a Qubit Fluorometer 2.0 (Thermo Fisher Scientific). All samples were sequenced on an Illumina HiSeq X Ten system at Macrogen (Korea) with the TruSeq DNA PCR-Free library kit (350 bp insert size) using 150 bp paired-end reads.

**Genome assembly**. WGS reads for *A. marmorata*, *A. megastoma*, and *A. obscura* were error-corrected and trimmed for adapters with "merTrim" from the Celera Assembler software v.8.3[95] (downloaded from the Concurrent Version System repository on 21 June 2017) using a k-mer size of 22 and the Illumina adapters option[96]. Celera Assembler was run with the following options: merThreshold = 0, merDistinct = 0.9995, merTotal = 0.995, unitigger = bogart, doOBT = 0, doToggle = 0; default settings were used for all other parameters. After assembly, the reads were mapped back to the assemblies using BWA MEM v.0.7.12, and their consensus was recalled using Pilon v.1.22[97]. The completeness of the three different assemblies was assessed with BUSCO v.3.0.1[98] based on the vertebrate gene set. Mitochondrial genomes were assembled separately from reads mapping to mitochondrial queries with the iterative MITObim v.1.8 approach[99] based on the MIRA v.4.0.2 assembler[100].

**Analysis of mitochondrial haplotypes**. RAD-sequencing reads mapping to the mitochondrial genome were converted to FASTA format using SAMtools v.1.3, BCFtools, and Seqtk v.1.0 (https://github.com/lh3/seqtk). Sequences corresponding to regions 10,630–10,720 and 12,015–12,105 of the *A. japonica* mitochondrial genome were aligned with default settings in MAFFT v.7.397[101] and the two resulting alignments were concatenated. The genealogy of mitochondrial haplotypes was reconstructed based on the GTRCAT substitution model in RAxML v.8.2.11[102] and used jointly with the concatenated alignment to produce a haplotype-genealogy graph with the software Fitchi v.1.1.4[103].

**Species-tree inference**. To estimate a time-calibrated species tree for the seven sampled *Anguilla* species, we applied the Bayesian molecular-clock approach of Stange et al.[31] to a subset of the dataset of 619,353 SNPs, containing data for the maximally five individuals per species with the lowest proportions of missing data (28 individuals in total: 1 *A. mossambica*, 3 *A. interioris*, 4 *A. bicolor*, and 5 of each remaining species). By employing the SNAPP v.1.3[32] package for the program BEAST 2 v.2.5.0[104], the approach of Stange et al.[31] integrates over all possible trees at each SNP and therefore allows accurate phylogenetic inference in the presence of incomplete lineage sorting. As the SNAPP model assumes a single rate of evolution for all substitution types, all SNAPP analyses were conducted separately for transitions and transversions. A maximum of 5000 SNPs was used in both cases to reduce run times of the computationally demanding SNAPP analyses. After exploratory analyses unambiguously supported a position of *A. mossambica* outside of the other six sampled anguillid species, the root of the species tree was calibrated according to published estimates for the divergence time of *A. mossambica*. Specifically, we constrained this divergence to 13.76 Ma (with a standard deviation of 0.1 myr), as reported by Jacobsen et al.[14] based on mitochondrial genomes of 15 anguillid species and three outgroup species. A justification of this

timeline is given in Supplementary Note 3. Five replicate Markov-chain Monte Carlo (MCMC) analyses were conducted and convergence was confirmed with effective sample sizes greater than 200, measured with the software Tracer v.1.7[105]. The posterior distributions of run replicates were merged after discarding the first 10% of each MCMC as burn-in, and maximum-clade-credibility (MCC) trees with node heights set to mean age estimates were generated with TreeAnnotator[106]. The robustness of divergence-time estimates was tested in a series of additional analyses, in which (i) alternative topologies were specified to fix the position of *A. interioris* (see below), (ii) species with strong signals of past introgression, *A. luzonensis* and *A. interioris* (see below), were excluded, (iii) genome assemblies of *A. marmorata*, *A. obscura*, and *A. megastoma* were used in combination with sequences and age constraints from Musilova et al.[39], or (iv) mitochondrial sequences for the same three species were used jointly with sequences and age constraints from Rabosky et al.[38]. A full description of these additional analyses is presented in Supplementary Note 4.

The relationships among the seven sampled species *A. marmorata*, *A. luzonensis*, *A. bicolor*, *A. obscura*, *A. interioris*, *A. megastoma*, and *A. mossambica* were further investigated based on maximum likelihood, using the software IQ-TREE v.1.7-beta12[46] and the same 28 individuals as in SNAPP analyses. RAD loci were filtered to exclude those with completely missing sequences and those with fewer than 20 (19,276 loci) or more than 40 variable sites (1 locus). The resulting dataset contained sequences for 1360 loci with a total length of 393,708 bp and 0.18% of missing data (Supplementary Fig. 1). The maximum-likelihood phylogeny was estimated from this set of loci with IQ-TREE's edge-linked proportional-partition model that automatically selects the best-fitting substitution model for each locus. Node support was estimated with three separate measures: 1000 ultrafast bootstrap-approximation replicates[107] and gene- and site-specific concordance factors[47]. These two types of concordance factors quantify the percentage of loci and sites, respectively, that support a given branch, and thus are a useful complement to bootstrap-support values that are known to often overestimate confidence with phylogenomic data[108]. The phylogenetic analyses with IQ-TREE were repeated with a set of 43 individuals that included 5 individuals from each of the four *A. marmorata* populations.

**Assessing genomic variation among and within species.** Genome-wide variation was estimated based on the dataset of 155,896 SNPs, after excluding sites linked within 10-kb windows with $R^2 > 0.8$ (Supplementary Fig. 1). We performed PCA using smartpca in EIGENSOFT v.6.0.1[109], including the function "lsqproject" to account for missing data, and through model-based clustering using ADMIXTURE v.1.3[40]. Five replicates, each testing for one to eight clusters ($K$) and 10-fold cross-validation were performed.

The software fineRADstructure v.0.3.1[41] was used to infer genomic variation among individuals by clustering them according to similarity of their RAD haplotypes in a coancestry matrix. Haplotypes were exported using "populations" in Stacks (see above), additionally filtering for a minor allele frequency above 0.02 and a mean log-likelihood greater than −10.0. The script "Stacks2fineRAD.py"[41] was used to convert haplotypes of loci with maximally 20 variable sites to the fineRADstructure input format, resulting in a set of haplotypes for 65,912 RAD loci (Supplementary Fig. 1). The coancestry matrix was inferred using RADpainter, and the MCMC clustering algorithm in fineSTRUCTURE v.4[110] was used to infer clusters of shared ancestry, setting the number of burnin iterations to 100,000, the sample iterations to 100,000, and the thinning interval to 1000. Finally, to reflect the relationships within the coancestry matrix, the inferred clusters were arranged according to a tree inferred with fineSTRUCTURE, using 100,000 hill-climbing iterations and allowing all possible tree comparisons.

**Detecting contemporary hybridization.** Based on the results of morphological and genomic PCA (Fig. 1, Supplementary Figs. 3, 5), analyses with ADMIXTURE (Supplementary Fig. 7) and fineRADstructure (Supplementary Fig. 8), and previous reports[18,111], we suspected that our dataset included recent hybrids between four species pairs: *A. marmorata* and *A. megastoma*, *A. marmorata* and *A. obscura*, *A. megastoma* and *A. obscura*, and *A. marmorata* and *A. interioris*. To verify these putative hybrids, we determined sites that were fixed in each of the four species pairs, considering only the "core"-group individuals for *A. marmorata*, *A. megastoma*, and *A. obscura* (see section "Morphological analyses"; 73, 26, and 26 individuals, respectively) and the three available individuals for *A. interioris* (Supplementary Table 1). At each fixed site for which no more than 20% of genotypes were missing, we then assessed the genotypes of the putative hybrids and plotted these in the form of "ancestry paintings"[42]. We expected that first-generation (F1) hybrids would be consistently heterozygous at nearly all sites fixed for different alleles between parental species (some few loci that appear fixed between the sampled individuals of the parental species might not be entirely fixed in those species), and that backcrossed individuals would show a heterozygosity of around 50% or less at these sites. For each verified F1 or backcrossed hybrid, we further quantified the proportion of its genome derived from the maternal species, $f_{m,genome}$, based on its genotypes at the sites fixed between parents and assuming that its mitochondrial genome reliably indicates the species of its mother. Finally, we also quantified the relative morphological similarity to the maternal species, $f_{m,morphology}$, for each hybrid, corresponding to the position of the hybrid on an axis connecting the mean morphology of the maternal species with the mean

morphology of the paternal species. Specifically we calculated this relative similarity as

$$f_{m,morphology} = 1 - \frac{1}{2}\left(\frac{PDH/TL - \overline{PDH/TL_m}}{\overline{PDH/TL_p} - \overline{PDH/TL_m}} + \frac{AD/TL - \overline{AD/TL_m}}{\overline{AD/TL_p} - \overline{AD/TL_m}}\right),$$

where $\overline{PDH/TL_m}$ is the mean PDH divided by TL of the maternal species, $\overline{PDH/TL_p}$ is the mean PDH divided by TL of the paternal species, $\overline{AD/TL_m}$ is the mean AD divided by TL of the maternal species, and $\overline{AD/TL_p}$ is the mean AD divided by TL of the paternal species.

**Detecting past introgression.** As our analyses of contemporary hybridization identified several backcrossed individuals, we assumed that, despite their old divergence times, tropical eel species may have remained connected by continuous or episodic gene flow. We thus tested for signals of past introgression among the seven species using multiple complementary approaches. Our first approach was motivated by the observation that *A. interioris* clustered with *A. marmorata* and *A. luzonensis* in the Bayesian species-tree analyses with SNAPP, but appeared as the sister to *A. bicolor* and *A. obscura* in the maximum-likelihood phylogeny generated with IQ-TREE, with strong support in both cases. Assuming that this discordance might have resulted from past introgression[112], we thus applied genealogy interrogation[48] to the dataset used for IQ-TREE analyses, composed of 1360 RAD loci with a total length of 393,708 bp. For each of these loci, we separately calculated the likelihood of three different topological hypotheses (H1–H3): *A. interioris* forming a monophyletic group with *A. marmorata* and *A. luzonensis* to the exclusion of *A. bicolor* and *A. obscura* (H1), *A. interioris* forming a monophyletic group with *A. bicolor* and *A. obscura* to the exclusion of *A. marmorata* and *A. luzonensis* (H2), or *A. marmorata*, *A. luzonensis*, *A. bicolor*, and *A. obscura* forming a monophyletic group to the exclusion of *A. interioris* (H3). These likelihood calculations were performed using IQ-TREE with the GTR substitution model, and two replicate analyses were conducted for each combination of locus and hypothesis. Per locus, we then compared the three resulting likelihoods and quantified the numbers of loci supporting H1 over H2, H2 over H1, H1 over H3, H3 over H1, H2 over H3, and H3 over H2. We expected that the true species-tree topology would be supported by the largest number of loci, and that introgression would, if present, increase the support for one of the alternative hypotheses relative to the other[52,113].

As a second approach for the detection of past introgression, we calculated Patterson's $D$ statistic[49,50] from biallelic SNPs included in the RAD-sequencing derived dataset of 619,353 SNPs (Supplementary Table 1). As this statistic is applicable to quartets of species in which one is the outgroup to all others and two species (labeled P1 and P2) are sister taxa, we calculated the $D$ statistic separately for all species quartets compatible with the species tree inferred through genealogy interrogation. In this species tree, *A. mossambica* is the sister to all other species and *A. interioris* is the sister to a clade formed by the two species pairs *A. marmorata* and *A. luzonensis* and *A. bicolor* and *A. obscura*. Per species quartet, the $D$ statistic was calculated as

$$D = (C_{ABBA} - C_{BABA})/(C_{ABBA} + C_{BABA}),$$

where $C_{ABBA}$ is the number of sites at which P2 and the third species (P3) share a derived allele and $C_{BABA}$ is the number of sites at which P1 and P3 share the derived allele. If sites were not fixed within species, allele frequencies were taken into account following Martin et al.[114]. In the absence of introgression, $D$ is expected to be zero; positive $D$ values are expected when introgression took place between P2 and P3, and negative $D$ values result from introgression between P1 and P3.

In addition to the above analyses based on RAD-sequencing-derived SNPs, the WGS data for *A. marmorata*, *A. megastoma*, and *A. obscura*, in combination with the available reference genome assembly for *A. anguilla*[57], allowed us to calculate $D$ statistics for this species quartet from a fully genomic dataset. To this end, WGS reads of the three species were mapped against the *A. anguilla* reference assembly using BWA MEM, and sorted and indexed using SAMtools. Duplicates were marked using Picard tools v.2.6.0 (http://broadinstitute.github.io/picard/), and indels were realigned using GATK v.3.4.64[115]. Per-species mean read coverage (71.31×, 64.80×, and 48.97× for *A. marmorata*, *A. megastoma*, and *A. obscura*, respectively) was calculated with bedtools v.2.26.0[116]. SNP calling was performed using SAMtools' "mpileup" command, requiring a minimum mapping quality (MQ) of 30 and a base quality (BQ) greater than 30, before extracting the consensus sequence using BCFtools v.1.6. The consensus sequences were converted to FASTQ format via SAMtools' "vcfutils" script for bases with a read depth (DP) between 15 and 140, and subsequently used to calculate the genome-wide $D$ statistic with *A. obscura* as P1, *A. marmorata* as P2, *A. megastoma* as P3, and *A. anguilla* as the outgroup.

The dataset of 619,353 RAD-sequencing-derived SNPs (Supplementary Table 1) was further used to calculated the $f_4$ statistic[51] as a separate measure of introgression signals, for the same species quartets as the $D$ statistic. The $f_4$ statistic is based on allele-frequency differences between the species pair formed by P1 and P2 and the species pair formed by P3 and the outgroup (as the $f_4$ statistic does not assume a rooted topology, P3 and the outgroup form a pair when P1 and P2 are monophyletic), and like the $D$ statistic, the $f_4$ statistic is expected to be zero in the absence of introgression. We calculated the $f_4$ statistic with the F4 program v.0.92[52]. As the

distribution of the $f_4$ statistic across the genome is usually not normally distributed, block-jackknife resampling is not an appropriate method to assess its significance; thus, we estimated p values based on coalescent simulations. These simulations were also conducted with the F4 program, internally employing fastsimcoal.v.2.5.2[117] to run each individual simulation. After a burnin period required to adjust settings for divergence times and population sizes in the simulations, the set of simulations allows the estimation of the p value for the hypothesis of no introgression as the proportion of simulations that resulted in an $f_4$ statistic as extreme or more extreme than the $f_4$ statistic of the empirical species quartet.

The genome-wide consensus sequences for *A. marmorata*, *A. megastoma*, and *A. obscura*, aligned to the *A. anguilla* reference genome assembly[57], were further used to test for introgressed regions on the largest scaffolds of the reference genome (11 scaffolds with lengths greater than 5 Mbp). To this end, maximum-likelihood phylogenies of the four species were generated with IQ-TREE for blocks of 20,000 bp, incremented by 10,000 bp, with IQ-TREE settings as described above for species-tree inference.

**Estimating effective population sizes**. Distributions of genome-wide coalescence times were inferred from WGS reads of *A. marmorata*, *A. megastoma*, and *A. obscura* using the pairwise sequentially Markovian coalescent model, implemented in the program PSMC v.0.6.4-r33[56]. Heterozygous sites were detected from consensus sequences in FASTQ format (see above) using the script "fq2psmcfa"[56], applying a window size of 20 bp (1.4% of windows contained more than one heterozygous site), and a scaffold-good-size of 10,000 bp. The PSMC analyses were run for 30 iterations, setting the initial effective population size to 15, the initial Θ to five, and the time-intervals option to "4 × 4 + 13 × 2 + 4 × 4 + 6", corresponding to 22 free parameters. To assess confidence intervals, 100 bootstrap replicates were performed using the script "splitfa"[56]. The PSMC plots were scaled using generation times reported by Jacoby et al.[118]; these were 12 years, 10 years, and 6 years for *A. marmorata*, *A. megastoma*, and *A. obscura*, respectively. Mutation rates were calculated based on pairwise genetic distances and divergence-time estimates inferred in our phylogenetic analyses. Uncorrected p-distances were 1.199% between *A. marmorata* and *A. megastoma*, 1.307% between *A. megastoma* and *A. obscura*, and 1.141% between *A. marmorata* and *A. obscura*. In combination with the divergence time of *A. megastoma* at 9.6954 Ma and the divergence time between *A. marmorata* and *A. obscura* at 7.2023 Ma, these distances resulted in mutation-rate estimates per site per generation of $r = 8.6 \times 10^{-9}$, $5.6 \times 10^{-9}$, and $5.2 \times 10^{-9}$ for *A. marmorata*, *A. megastoma*, and *A. obscura*, respectively.

**Identification of genomic rearrangements**. Structural genomic rearrangements among *A. marmorata*, *A. megastoma*, and *A. obscura* were identified by performing whole-genome alignment for the three newly generated genome assemblies and the two previously available genome assemblies for *A. japonica*[119] and *A. anguilla*[57]. Whole-genome alignments were generated in a pairwise manner with the program LASTZ v.1.04[120], aligning the assemblies of *A. japonica*, *A. marmorata*, *A. megastoma*, and *A. obscura* separately to the *A. anguilla* reference-genome assembly. Based on the orientation and order of alignment blocks, we first determined regions of potential inversions and rearrangements and then investigated each whole-genome alignment more specifically for the presence of rearrangements in those regions. We generated dot plots comparing pairs of scaffolds with potential rearrangements and used these plots to visually confirm the presence or absence of each rearrangement (Supplementary Table 11, Supplementary Fig. 21). To exclude assembly errors as a cause of false signals of rearrangements, we mapped the WGS reads of *A. marmorata*, *A. megastoma*, and *A. obscura* to the species-specific genome assemblies using BWA MEM, and investigated the distributions of mapped reads with and without proper mate pairing (Supplementary Figs. 21, 22). A full description of the methods used to identify and verify genomic rearrangements is provided in Supplementary Notes 5 and 6. We further applied gene prediction with AUGUSTUS v.3.3.3[63] to the *A. anguilla*[57] reference genome (Supplementary Note 7), allowing us to determine the locations of rearrangements relative to coding sequences. For rearrangements within genes, we applied TBLASTX searches[92] to determine the locations and orientations of regions homologous to exons of zebrafish (*Danio rerio*) genes (assembly version GRCz11; NCBI accession GCA_000002035.4[121]) (Supplementary Table 12).

**Identification of nonsynonymous substitutions**. To investigate possible causes of cytonuclear incompatibility between *A. marmorata* and *A. megastoma*, we analyzed both the RAD-sequencing-derived nuclear SNPs as well as mitochondrial genomes produced with WGS for substitutions that change the amino-acid sequence of proteins. For each of the 302 nuclear sites fixed between *A. marmorata* and *A. megastoma*, we determined whether it was localized within a coding sequence predicted with AUGUSTUS (Supplementary Note 7), and if so, whether it affects the amino-acid translation. For the resulting set of nine fixed sites localized within coding sequences (Supplementary Table 13), we used BLASTX searches[92] to identify homologous proteins in the zebrafish proteome. Mitochondrial gene sequences were identified from the newly generated mitochondrial genome assemblies of *A. marmorata* and *A. megastoma* using nucleotide sequences of the *A. anguilla* mitochondrial genome (NCBI accession NC_006531) as queries in TBLASTN searches[92]. The identified sequences were aligned separately for each

mitochondrial gene with MAFFT and translated into amino acids with the vertebrate mitochondrial genetic code (Supplementary Table 15).

**Reporting summary**. Further information on research design is available in the Nature Research Reporting Summary linked to this article.

## Data availability
The raw RADseq data are deposited on the NCBI SRA database with project number PRJNA590038. Genome assemblies and WGS reads for *A. marmorata*, *A. megastoma*, and *A. obscura* are deposited on ENA with project number PRJEB32187. Haplotype files, alignment files, SNP datasets in VCF format, and input and output of phylogenetic analyses are available from the associated Dryad repository (https://doi.org/10.5061/dryad.ncjsxksr1). Previously available datasets used in this study include the NCBI accessions CM002536, NC_006531, GCA_000002035.4, GCA_000695075, GCA_000470695, NM_001020485, and DQ649453. The source data underlying Figs. 1b, c, 2a–d, i–l, 3c, d, and Supplementary Figs. 3–5, 7, and 15–18 are provided as a Source Data file.

## Code availability
Code for computational analyses is available from Github (http://github.com/mmatschiner/anguilla).

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

## Acknowledgements

Funding for this study was provided by the Austrian Science Fund (FWF, project P28381-B29 to R. Schabetsberger) and the Norwegian Research Council (FRIPRO project 275869 to M.M.). We thank Anthony Acou, David Boseto, Donna Kalfatak, Rilloy Leaana, Finn Økland, Christine Pöllabauer, Alexander Scheck, Ursula Sichrowsky, and Meelis Tambets for assistance with field work, Franz Gassner for help with data analysis, Ian Goodhead for his support in the laboratory, and Milan Malinsky for advice on whole-genome alignment. We further thank Olaf L. F. Weyl and acknowledge the DST/NRF South African Research Chair in Inland Fisheries and Freshwater Ecology (Grant No. 110507) and the NRF-SAIAB Collections Platform for the provision of genetic tissue samples. We thank the governments of American Samoa, France, Indonesia, Papua New Guinea, Philippines, Samoa, Solomon Islands, South Africa, Swaziland, Taiwan, and Vanuatu for issuing sampling permits. All computational work was performed on the Abel Supercomputing Cluster (Norwegian Metacenter for High-Performance Computing (NOTUR) and the University of Oslo), operated by the Research Computing Services group at USIT, the University of Oslo IT Department. Peter Comes provided valuable comments that allowed us to improve the manuscript.

## Author contributions

R. Schabetsberger and R.J. conceived the project. R. Schabetsberger, R.J., C.G., M.M., J.M. I.B., and R. Sommaruga planned and oversaw the project. R. Schabetsberger, C.G., Y.-S. H., and E.F. contributed specimens. C.G., R.J., and R. Schabetsberger organized RAD sequencing. S.W. performed morphological analyses. J.M.I.B. and B.E. prepared genomic datasets. O.K.T. performed genome assembly. M.M. and J.M.I.B. performed population genomic and phylogenomic analyses. M.M. performed analyses of structural rearrangements. M.M. and J.M.I.B. prepared the figures. M.M. and J.M.I.B. wrote the manuscript with input from all authors.

## Competing interests

The authors declare no competing interests.
