## [Peer Review File · Nature Communications]

Reviewers' Comments:

Reviewer #1:

Remarks to the Author:

This manuscript confronts an iconic issue in evolutionary biology – the population structure of freshwater eels of the genus *Anguilla*, which is shaped by their intriguing life cycles. The authors have collected a very extensive state-of-the-art genomic dataset, with which they convincingly demonstrate that different eel species resist genomic homogenization, despite forming abundant hybrids. The responsible mechanisms remain less clear.

This study joins the growing body of work applying novel genomics possibilities to the age-old biological enigmas presented by eel biology. The paper is of immediate interest to anyone studying eel biology or population genetics in general. Furthermore, the data will conceivably support numerous follow-up studies.

The study relies on a large amount of data, which are processed using sophisticated methods. Its narrative is therefore often quite technical. I think the paper could be made more interesting for a general audience if the authors 'zoomed out' now and then, to emphasize the general principles at play and how these interact with eel biology.

For example, one of the main conclusions of the paper is that large-scale introgression is hindered by large effective population sizes. Here, it would be interesting to reflect on eels' migratory behavior and associated panmixia, which present the paradox of both encouraging hybridization (because of few and shared spawning locations) and enforcing species boundaries through stringent purifying selection.

Similarly, the data holds a lot of promise for the study of (eel) speciation. It invites constructing and evaluating scenarios in which eels ancestrally share spawning grounds, yet diverge genetically; or alternatively, come to share spawning grounds after speciation; or diverge because of new spawning grounds (see my point 2 below).

There are several issues that I would appreciate the authors' perspectives on:

1. The Indo-Pacific area is home to more *Anguilla* species than analyzed here. These other species almost certainly also engage in wide-spread hybridization. Do you see signatures of genomic variation that can be attributed to 'cryptic' populations? Would the existence of these affect any of your analyses?
2. The data show that a large fraction of the overall genomic variation uncovered resides within the *A. marmorata* genomes (fig 1c, S5). This variation correlates with the four populations for this species (fig S4a). These populations appear highly distinct, with perhaps one hybrid individual between Western South Pacific and South China Sea (fig S4a, not mentioned in the text - the gene flow between *A. marmorata* populations is therefore less than between *A. marmorata* and other species!) Why did you then only include one population in the phylogeny reconstructions (individuals listed in fig S17)? This is apparently also the population (Western South Pacific) that is closest to the other species (fig S5). Would a phylogeny using another, or all four populations present different topologies?
3. Related to this, the data show some patterns of genomic variation that are not discussed in the text, but that I think are relevant. Fig S4e appears to show some structure in the *A. obscura* species, does PC1 here correlate with geography (SAW, VAG)? Do the mtDNA haplotypes (fig S2) correlate

with biology or geography (especially for *A. marmorata* and *A. obscura*)?

4. But most interestingly, the PCA plots (fig 1c, S5) clearly to show that individuals within all species appear to be ordered on 'trajectories'. Can these be matched to relevant biology (similar to pseudotime trajectories in single-cell developmental biology studies)?

5. The low frequency of backcrosses, and the absence of later generation backcrosses or F2 hybrids are used to support the argument against large-scale population mixing. However, these frequencies may not be all that low and the absences not surprising, given the sample size. With hundreds of individuals, and a few percent F1 hybrids, F2 hybrids could already be too rare to spot. This is of course a naïve calculation, given the existence of hybridization hotspots (e.g. VAG, table S8). However, F1 hybrid frequencies here cannot be easily extrapolated to F2 without knowing the general contribution of VAG eels to next generations. In other words, migration success itself could affect hybridization. (PS individual VAG13087 has a large $f[m, genome]$ for a backcross, could it be a later-generation hybrid?)

6. You hypothesize that cytonuclear incompatibilities could contribute to species barriers. I think the genome sequencing (WGS) data generated could actually be used to look for a mechanistic basis of these, by aligning contigs to the *A. anguilla* reference and scanning for species-specific structural mapping inconsistencies.

7. Figs S15 and S16 often appear to show groupings based on morphological characters, but these groups are probably all just based on overall body size. Can these measurements be normalized for length/weight? Or age (did you collect /analyze otoliths)?

8. Fig. 2a/e: Not all hybrids are marked as such in fig. 2e?

Reviewer #2:

Remarks to the Author:

This manuscript reports on the distribution of genetic variation among seven *Anguilla* species, which have high potential for contemporary gene exchange given the cooccurrence of several species during the breeding phase of their life cycle. The study includes a wealth of genomic data that appear to have been handled with proper care in terms of bioinformatics. The data were thoroughly analyzed statistically to address questions about species recognition, contemporary hybridization, and introgression deeper in the history of evolution of the genus. The figures are impressive and contain a large amount of information from various analyses.

This is principally a case study of how species boundaries persist despite evidence for both contemporary and historical hybridization. This case study joins other studies of plants and animals (including other fishes), that have shown with genomic data that species boundaries and impressive biological diversity persist despite recent and more ancient hybridization. Consequently, the findings and interpretations do not shift my thinking about the challenge of hybridization to the maintenance of species. The framing of the novelty of the questions, and the gap in our knowledge, in the introduction goes beyond how the manuscript presents the work in the discussion. In the discussion the findings are interpreted principally in the context of *Anguilla* evolution, which seems more appropriate.

The research is sound and supports the fairly established view that species can evolve and maintain trait differences despite genetic exchange throughout their history. Genomic data are bringing new forms of evidence and analyses to support this view, but is not new or controversial in evolutionary

biology.

Reviewer #3:

Remarks to the Author:

The authors present a genomic analysis of *Anguilla* eels, connecting evidence from contemporary examples of hybridization with historical introgression, despite significant divergence time estimates between the species. The role of hybridization and introgression to the topic of speciation is of keen importance, and rarely are contemporary instances of hybridization considered alongside robust estimates of historical introgression.

To determine contemporary rates of hybridization, the authors used genomic and morphological analyses to quantify different hybrid classes. Their genomic data included a mix of reduced representation sequencing, as well as whole genome resequencing. They were able to successfully identify several hybrids, and this made up a surprising portion of the sampled individuals. To estimate historical level of possible introgression, the authors used a range of robust phylogenomic tools. Fairly convincingly showing that variation in the topology amongst these trees is the result of historical introgression.

I found the writing clear and the flow of the manuscript good, although sometimes ending up further into the weeds than I would expect in broad journal like this (e.g. lines 150-191). While I do have some suggested improvements, I think the paper will be of general interest to the readers of *Nature Communications* and am supportive of publication.

Even more importantly, I might request the authors make the figures for some of my own papers— what a breath of fresh air it is to see clear, informative, and thoughtful figures, with a very appropriate use of color (i.e. beautiful evidence!).

Specific Comments:

Figure 1a – Regarding the positional information, it seems odd to just use the GBIF data, and not include your own collection information. For example, the SAW site obviously has lots of obscura, but that wasn't included in the GBIF dataset, so doesn't show up on the map. I might recommend combining the datasets for the locality information?

Line 77-78 – Given the overlap in morphospace, it is somewhat difficult to see "intermediate" individuals here? Might be worth revising this sentence.

Figure 2e-f – I guess here I am a bit surprised that the multivariate analysis is showing morphology, and not the genomic analysis, given the genomic analysis has more discriminatory power to cluster the groups.

Figure 2i – I appreciate the ancestry painting and the multivariate genomic analysis. I wonder if the authors had also considered using "triangle plot" analyses (e.g. DOI: <https://doi.org/10.1098/rspb.2017.2081>) to identify different hybrid classes?

Lines 162 – 191 – I guess I am not overly surprised by the fact that the putative hybrids are intermediate in size (as would be predicted from a multi-locus quantitative trait, and has been demonstrated quite a few times before in hybrid zones). I also think the transgressive piece here, while interesting, seems somewhat tangential to your main question. Therefore, I might suggest you reduce the emphasis in the text on this morphological analysis, at least here.

Figures 3c and d – I was always told not to include a regression line if the relationship was not significant..

Lines 301-302 – On my first reading, this made me think you were suggesting that the production of adult F1 hybrids don't also speak to the possible post-mating barriers. I appreciate the point following from your question about whether barriers that act after the F1 are very important, but I would rephrase this slightly.

Please find below our replies to all reviewers' comments (comments in *italics*, our replies in regular font):

Reviewer #1

*This manuscript confronts an iconic issue in evolutionary biology – the population structure of freshwater eels of the genus *Anguilla*, which is shaped by their intriguing life cycles. The authors have collected a very extensive state-of-the-art genomic dataset, with which they convincingly demonstrate that different eel species resist genomic homogenization, despite forming abundant hybrids. The responsible mechanisms remain less clear.*

We thank Reviewer 1 for reviewing our manuscript and for providing constructive suggestions. Following the suggestions of Reviewer 1, we have now added further analyses that allowed us to shed more light on the mechanisms through which tropical eel species resist genomic homogenization, particularly through cytonuclear incompatibility. These additional analyses and results are described below.

This study joins the growing body of work applying novel genomics possibilities to the age-old biological enigmas presented by eel biology. The paper is of immediate interest to anyone studying eel biology or population genetics in general. Furthermore, the data will conceivably support numerous follow-up studies.

The study relies on a large amount of data, which are processed using sophisticated methods. Its narrative is therefore often quite technical. I think the paper could be made more interesting for a general audience if the authors 'zoomed out' now and then, to emphasize the general principles at play and how these interact with eel biology.

We appreciate both of these suggestions to 'zoom out' and to reduce the technical jargon of the manuscript, and followed these in our revision of the manuscript. We have shortened multiple technical sections, removed some details from the main text that are also included in figures or the Supplementary Information, and rephrased complicated sections. In particular, we made the description of morphological results in the Results section "High frequency of contemporary hybridization" (lines 150-191 in the original manuscript, now lines 144-163) more concise, a change that was also suggested by Reviewer 3. We also followed the suggestions of Reviewer 1 by extending the interpretation of our results in various parts of the Results and Discussion sections, providing more context about the biology of tropical eels and how it is relevant for our conclusions (see our replies to the next comments).

For example, one of the main conclusions of the paper is that large-scale introgression is hindered by large effective population sizes. Here, it would be interesting to reflect on eels'

migratory behavior and associated panmixia, which present the paradox of both encouraging hybridization (because of few and shared spawning locations) and enforcing species boundaries through stringent purifying selection.

We agree that the panmixia of eel populations is indeed worth emphasizing in relation to the efficiency of purifying selection. In our revised manuscript, we have added a statement to the Discussion section to highlight its relevance (lines 339-341).

Similarly, the data holds a lot of promise for the study of (eel) speciation. It invites constructing and evaluating scenarios in which eels ancestrally share spawning grounds, yet diverge genetically; or alternatively, come to share spawning grounds after speciation; or diverge because of new spawning grounds (see my point 2 below).

We agree that a brief discussion of speciation scenarios is warranted and have now included one in the Discussion of the revised manuscript. We also outline how population-level whole-genome resequencing may be used in future studies to shed more light on past and ongoing speciation processes (lines 344-357).

There are several issues that I would appreciate the authors' perspectives on:

*1. The Indo-Pacific area is home to more *Anguilla* species than analyzed here. These other species almost certainly also engage in wide-spread hybridization. Do you see signatures of genomic variation that can be attributed to 'cryptic' populations? Would the existence of these affect any of your analyses?*

It is correct that further *Anguilla* species are known from the Indo-Pacific and that hybridization with these species could be expected if they do co-occur locally. We mention in the Results section "Extensive sampling" that four Indo-Pacific eel species are not included in our dataset: *A. celebesensis*, *A. bengalensis*, *A. borneensis*, and *A. reinhardtii* (lines 68-70). Mitochondrial sequences of these species are available on NCBI and would have allowed their identification if any individuals had possessed mitochondrial genomes of these species. We are also confident that hybrids between sampled and unsampled species could have been identified from nuclear data if they had been included in our dataset, due to their expected increased heterozygosity, their position in PCA analyses, and their signals in the ADMIXTURE and fineRADstructure analyses. To clarify this aspect, we have now added a new paragraph to the end of the Results section "High frequency of contemporary hybridization" (lines 164-171).

*2. The data show that a large fraction of the overall genomic variation uncovered resides within the *A. marmorata* genomes (fig 1c, S5). This variation correlates with the four populations for this species (fig S4a). These populations appear highly distinct, with perhaps one hybrid*

individual between Western South Pacific and South China Sea (fig S4a, not mentioned in the text - the gene flow between A. marmorata populations is therefore less than between A. marmorata and other species!) Why did you then only include one population in the phylogeny reconstructions (individuals listed in fig S17)? This is apparently also the population (Western South Pacific) that is closest to the other species (fig S5). Would a phylogeny using another, or all four populations present different topologies?

While *A. marmorata* was the only species for which we detected within-species population structure, the genomic variation residing in *A. marmorata* is probably not as large as it might appear from the PCA plots shown in Fig. 1c and Supplementary Figure 5. This is because in these PCA plots, individuals are colored according to their mitochondrial genomes, which means that almost all hybrids (all individual labelled in Fig. 1c) were marked in the color of *A. marmorata*. Without these hybrid individuals, the range in PC1 and PC2 occupied by *A. marmorata* individuals is comparable or even lower than the ranges occupied by *A. megastoma* and *A. obscura* individuals. The separation between the South China Sea population and the three other *A. marmorata* populations in fact dominates PC4 and PC3 in Supplementary Figure 5b and d, respectively, and is visible on PC1 in Fig. 1c and Supplementary Figure 5a,c; however, this pattern could in part be due to the much larger sample size for *A. marmorata* compared to the other species, which likely influenced the orientation of the PC axes. To test this assumption, we repeated PCA with a balanced set of three to four individuals per species (excluding all hybrids and *A. mossambica* of which we only had a single sample); this PCA is shown in the figure below. As expected, axis orientation now shows the separation of *A. megastoma*, the most divergent of the six species on PC1, and PC2-4 separate the remaining species. The strongest within-species separation is nevertheless found in *A. marmorata*, where the South China Sea individual is separated from its conspecifics on PC3.

We agree that the apparent absence of among-population hybrids is noteworthy. The one individual that appears intermediate between the western South Pacific and the South China Sea populations in Supplementary Figure 4a is VAG12033, which is characterized by one of the highest proportions of missing data. With only 725,846 sequencing reads available for this individual (Supplementary Table 1), it is only slightly above the absolute minimum of 600,000 that we imposed for individuals to be included in our analyses, and far below the mean number of reads among all included individuals (mean = 4,858,017 reads). We thus assume that the outlier position of this individual in the PCA is more likely caused by its high proportion of missing data than by admixture between the two populations. In the revised manuscript, we have now marked this individual in Supplementary Figure 4 and mention in the figure legend that it is characterized by a high proportion of missing data. We also added a statement to point out the absence of among-population hybrids in *A. marmorata*: “In contrast to these signals of interspecific hybridization, no *A. marmorata* individuals had genotypes clearly intermediate between the four distinct populations within the species (Supplementary Figure 4).”

For the phylogeny reconstruction with IQ-TREE shown in Supplementary Figure 19a (Supplementary Figure 17 in the original manuscript) as well as the reconstructions with SNAPP shown in Fig. 1d and Supplementary Figure 6a-h, we strictly selected the maximally five individuals per species with the lowest proportion of missing data; thus, the fact that all five individuals selected for *A. marmorata* are from the Western South Pacific was a coincidence. Given that the model implemented in SNAPP assumes panmictic populations, this coincidence was in fact beneficial because the use of *A. marmorata* individuals from multiple populations would have increased the violation of the SNAPP model. However, we agree that it would have been useful to include *A. marmorata* individuals from the other three populations in IQ-TREE

analyses. We thus now performed additional IQ-TREE analyses with five *A. marmorata* individuals from each of the four populations from this species, again selecting those with the lowest proportions of missing data per population or species. These analyses showed the same among-species relationships as the earlier IQ-TREE analyses, again with full bootstrap support for each species split (see the new panel b of Supplementary Figure 19). All *A. marmorata* individuals formed one monophyletic group. The relationships inferred among *A. marmorata* individuals are consistent with the haplotype-genealogy graph (Supplementary Figure 2), the genomic PCA (Supplementary Figure 4), and the results of the ADMIXTURE and fineRADstructure analyses (Supplementary Figures 7-8): The most distinct population within *A. marmorata* is the one from the South China Sea and weaker separation is found among the other three populations.

Despite the consistent support for the separation of the South China Sea population from the other *A. marmorata* populations and the apparent signal for coancestry between individuals of the South China Sea population and the geographically close species *A. luzonensis*, we note that introgression statistics do not support gene flow between these groups (Supplementary Table 10).

3. Related to this, the data show some patterns of genomic variation that are not discussed in the text, but that I think are relevant. Fig S4e appears to show some structure in the A. obscura species, does PC1 here correlate with geography (SAW, VAG)? Do the mtDNA haplotypes (fig S2) correlate with biology or geography (especially for A. marmorata and A. obscura)?

We had noticed the apparent clustering of *A. obscura* individuals along PC1, but did not discuss it because initial tests had not shown any correlation between positions along PC1 and sampling sites. This comment of Reviewer 1 now led us to investigate possible correlations in more detail. We compared the positions along PC1, 2, 3, and 4 with sampling sites (shown in panels a and b in the below figure), sampling year (panels c and d), terminal length (TL; panels e and f), and the proportion of missing data (not shown). None of our comparisons indicated correlations; therefore, we do not further discuss the clustering of *A. obscura* individuals. This statement has been added to the legend of Supplementary Figure 4.

The population structure in *A. marmorata* revealed by PCA of nuclear SNPs is consistent with the haplotype genealogy graph: Within *A. marmorata*, haplotypes from the South China Sea population are largely separated (F_{ST} between 0.59 and 0.71) from those of the other populations, even though the overall most frequent haplotype is also observed in the South China Sea population. In contrast, haplotypes of the three populations from the western Indian Ocean, Java, and the western South Pacific are not separated by population (F_{ST} between 0.02 and 0.03). Like in the PCA, *A. obscura* individuals also do not cluster by geography in the mitochondrial haplotype genealogy graph ($F_{ST} = 0$ between SAW and VAG). In the revised manuscript, we modified Supplementary Figure 2 to show this information and changed the legend of Supplementary Figure 2 accordingly.

4. But most interestingly, the PCA plots (fig 1c, S5) clearly show that individuals within all species appear to be ordered on ‘trajectories’. Can these be matched to relevant biology (similar to pseudotime trajectories in single-cell developmental biology studies)?

We apologize for leaving the apparent ‘trajectories’ in PCA plots uncommented in the original manuscript. These patterns could have been relevant for our conclusions as they could in principle indicate different admixture proportions among individuals. However, as the figure below reveals, these patterns are not caused by any interesting biological properties but instead represent an artefact resulting from different proportions of missing data among individuals. In the two panels of the figure, we compare the positions of *A. megastoma* (left panel) and *A. obscura* (right panel; these are the two species with the most obvious ‘trajectories’ in Supplementary Figure 5c) individuals with the numbers of missing genotypes per individual, and find that the correlation is significant (linear regression; $p < 0.001$) in both cases and that the individuals with the greatest proportions of missing data are those that appear at the right end of the ‘trajectories’ in Supplementary Figure 5c. The influence of missing data on PCA has previously been known and was discussed by Patterson et al. (2006; Population structure and eigenanalysis. PLOS Genetics, 2, e190). By using the function “lsqproject” of the EIGENSOFT smartPCA program, we aimed to reduce the influence of missing data; however, as the significant correlations in the figure below show, the function apparently could not remove this influence entirely. In the revised manuscript, we included the below figure as panels e) and f) in Supplementary Figure 5 and discuss its significance briefly in

the figure legend.

5. The low frequency of backcrosses, and the absence of later generation backcrosses or F2 hybrids are used to support the argument against large-scale population mixing. However, these frequencies may not be all that low and the absences not surprising, given the sample size. With hundreds of individuals, and a few percent F1 hybrids, F2 hybrids could already be too rare to spot. This is of course a naïve calculation, given the existence of hybridization hotspots (e.g. VAG, table S8). However, F1 hybrid frequencies here cannot be easily extrapolated to F2 without knowing the general contribution of VAG eels to next generations. In other words, migration success itself could affect hybridization. (PS individual VAG13087 has a large $f[m,genome]$ for a backcross, could it be a later-generation hybrid?)

For hybridization to lead to introgression, which could ultimately destabilize species boundaries, repeated backcrossing into the same parental species would be required. This could be evidenced by individuals with $f_{m,genome}$ values between 0 and 0.25 or between 0.75 and 1; however, no such values were observed in our dataset, except for some very close to 0.75 and 1: Among all 210 *A. marmorata*, *A. megastoma*, or *A. obscura* individuals that were not included in the “core” groups, had not already been identified as hybrids, and had over 2 million sequencing reads, none had an $f_{m,genome}$ value below 0.977 or a heterozygosity at fixed sites above 0.040. This information was missing from the original manuscript but has now been added to Supplementary Table 7 of the revised manuscript. These values are roughly as expected for sixth-generation backcrosses but could also result for unadmixed individuals due to uncertainty in the determination of fixed sites between species. Either way, the gap in the $f_{m,genome}$ distribution between 0.767 and 0.977 indicates the absence of third-, fourth-, and fifth-generation backcrosses in our dataset.

However, we do not claim that later-generation hybrids are entirely absent from the population, only that they seem to occur at much lower (if any) frequency than F1 hybrids, indicating a loss in fitness in later-generation backcrosses that can stabilize species boundaries. We agree with Reviewer 1 that a lower frequency of F2 hybrids compared to F1 hybrids would not in itself be evidence for decreased fitness because it could also reflect the rarity of encounters between two F1 hybrids. However, this argument does not apply to the frequency of backcrosses, as encounters between F1 hybrids and members of the parental species are not expected to be rare (our sampling shows that they do co-occur). Thus, given that our sample size was sufficient to identify F1 hybrids, we expect that it would also have been sufficient to identify first- and later-generation backcrosses and the fact that these are observed only at lower frequency or not at all indicates their decreased fitness. To clarify that it is only the low frequency of backcrosses (and not that of F2 hybrids) that indicates the decline of fitness, we have removed the unnecessary mention of the absence of F2 hybrids from our revised manuscript.

The individual VAG13087 unfortunately has a large proportion of missing genotypes at the 302 sites that are fixed differently between *A. marmorata* and *A. megastoma*; 177 genotypes are missing and only 125 genotypes are available. The individual is heterozygous at 73 of the 302 fixed sites, homozygous for the *A. megastoma* allele at 48 of these sites, and homozygous for the *A. marmorata* allele at 4 of the sites (note that a small number of homozygous sites shared with both parents is not unexpected and may result from uncertainties in the determination of fixed sites; even the clearest examples of F1 hybrids, such as SAW17B49, share up to 8 homozygous sites with parental species). Its $f_{m,genome}$ is therefore $(73/2+4) / 125 = 0.324$. This could in principle be explained if VAG13087 was the offspring of one backcross and one offspring of a backcross and an F1 hybrid; in that case, its $f_{m,genome}$ would be expected to be 0.3125. However, an $f_{m,genome}$ value between 0.25 and 0.5 could not be explained by repeated backcrossing with the same paternal species, the type of backcrossing that can lead to introgression. Thus, if VAG13087 should be the offspring of backcrosses and F1 hybrids, this would not affect our conclusions regarding the stable species boundaries among tropical eel species. In the revised manuscript, we added a note to Supplementary Table 7 (in which $f_{m,genome}$ is reported), explaining that the $f_{m,genome}$ value of VAG13087 could indicate earlier admixture:

“The $f_{m,genome}$ between 0.25 and 0.5 could indicate that one of the grandparents of VAG13087 was itself admixed. However, as VAG13087 has a comparatively large proportion of missing data and its genotypes are available for only 125 of 302 fixed sites, we do not consider this evidence of earlier admixture reliable enough to warrant further discussion.”

6. You hypothesize that cytonuclear incompatibilities could contribute to species barriers. I think the genome sequencing (WGS) data generated could actually be used to look for a mechanistic basis of these, by aligning contigs to the *A. anguilla* reference and scanning for species-specific structural mapping inconsistencies.

We thank Reviewer 1 for this excellent suggestion, which we have followed and extended. In brief, we have performed whole-genome alignment for *A. marmorata*, *A. megastoma*, *A. obscura*, and *A. japonica* (using the published assembly for this species) to the reference genome assembly of *A. anguilla*, and developed and applied a thorough pipeline for the detection of genomic rearrangements from these whole-genome alignments. This approach is described concisely in the new Methods section “Identification of genomic rearrangements” in the main text and more details are provided in the new Supplementary Notes 5-7. We identified several clear examples of genomic rearrangements, including an inversion within the *myhc4* gene (encoding myosin heavy chain 4) that appears to be present heterozygously in *A. megastoma*. These results are presented in a whole new Results section in the main text entitled “Cytonuclear incompatibility between *A. marmorata* and *A. megastoma*”, the accompanying new Figure 4, the new Supplementary Tables 11 and 12, and the new Supplementary Figures 21 and 22.

Going beyond the suggestion of Reviewer 1, we also investigated our nuclear RAD-sequencing data and the mitochondrial genomes of *A. marmorata* and *A. megastoma* for non-synonymous differences between the species. These analyses, which are described in the new Methods section “Identification of non-synonymous substitutions”, identified a fixed non-synonymous substitution in an exon of the nuclear gene *ttna* (encoding titin) and revealed a substantially increased density of non-synonymous substitutions in *atp6* compared to other mitochondrial genes. These results are also presented in the new Figure 4 and in Supplementary Tables 13-15, and we interpret our findings in the new Results section “Cytonuclear incompatibility between *A. marmorata* and *A. megastoma*” together with the identified genomic rearrangements. The proteins encoded by the three genes with marked differences between *A. marmorata* and *A. megastoma* (myosin heavy chain, titin, and ATP synthase) are all linked to muscle function and could thus play a role in migration fitness. Notably, all three genes have also previously been identified in connection to cytonuclear incompatibility between the two Atlantic eel species *A. anguilla* and *A. rostrata* (Gagnaire et al. 2012; Jacobsen et al. 2017). We thus speculate that similarly to the Atlantic species pair, cytonuclear incompatibilities may also be caused by these genes between *A. marmorata* and *A. megastoma*.

7. Figs S15 and S16 often appear to show groupings based on morphological characters, but these groups are probably all just based on overall body size. Can these measurements be normalized for length/weight? Or age (did you collect /analyze otoliths)?

Unfortunately, otoliths were not collected and only fragmentary and indirect age information is available for the individuals in our dataset (through body size and in some cases through a record indicating whether these were silver eels). Therefore, we now added plots for all measurements scaled by terminal length, as panels m-v) in Supplementary Figures 17 and 18 of the revised manuscript. These plots (Supplementary Figure 17q) show transgression in one measurement (eye distance standardized by terminal length), in one of the two hybrid individuals (VAG12044) that are also transgressive for terminal size. These plots also show more clearly the difference in standardized predorsal length between *A. marmorata* and *A. obscura* (Supplementary Figure 18n).

8. Fig. 2a/e: Not all hybrids are marked as such in fig. 2e?

It is correct that not all hybrids were shown in Fig. 2e; five hybrids from American Samoa, two hybrids from Samoa, and one hybrid from Vanuatu were not included in Fig. 2e. The reason for this omission was that the displayed morphological measurements (PDH/TL and AD/TL) were not available for these individuals (Supplementary Table 1). To clarify this, we have now added the following sentence to the figure legend: “Hybrids identified in a) are marked with specimen IDs, excluding hybrids from Samoa (SAW) and American Samoa (SAA) and one hybrid from Vanuatu (VAG; VAG13087) for which the displayed measurements were not available (Supplementary Table 1).” Note, however, that in response to a comment of Reviewer 3, we have now placed the former panels e-h of Figure 2 in the Supplementary Information as a separate new figure, Supplementary Figure 16. We have replaced these panels in Figure 2 of the revised manuscript with similar plots showing genomic instead of morphological variation (now Fig. 2a-d).

Reviewer #2

*This manuscript reports on the distribution of genetic variation among seven *Anguilla* species, which have high potential for contemporary gene exchange given the cooccurrence of several species during the breeding phase of their life cycle. The study includes a wealth of genomic data that appear to have been handled with proper care in terms of bioinformatics. The data were thoroughly analyzed statistically to address questions about species recognition,*

contemporary hybridization, and introgression deeper in the history of evolution of the genus. The figures are impressive and contain a large amount of information from various analyses.

We thank Reviewer 2 for reviewing our manuscript and for appreciating the thoroughness of our analyses and the design of our figures.

This is principally a case study of how species boundaries persist despite evidence for both contemporary and historical hybridization. This case study joins other studies of plants and animals (including other fishes), that have shown with genomic data that species boundaries and impressive biological diversity persist despite recent and more ancient hybridization. Consequently, the findings and interpretations do not shift my thinking about the challenge of hybridization to the maintenance of species. The framing of the novelty of the questions, and the gap in our knowledge, in the introduction goes beyond how the manuscript presents the work in the discussion. In the discussion the findings are interpreted principally in the context of Anguilla evolution, which seems more appropriate.

The research is sound and supports the fairly established view that species can evolve and maintain trait differences despite genetic exchange throughout their history. Genomic data are bringing new forms of evidence and analyses to support this view, but is not new or controversial in evolutionary biology.

We agree that the observation of interspecific genetic exchange has already become fairly established, but we feel that there is still a lot to learn about the details of how long genetic exchange can persist and how genome homogenization can be avoided between hybridizing species. We believe that our study contributes to an improved understanding of these aspects and are therefore excited to share our findings with the field. We hope that Reviewer 2 will find our changes and additions to the revised manuscript, revealing possible causes of cytonuclear incompatibility between *A. marmorata* and *A. megastoma*, of greater novelty than our original manuscript.

Reviewer #3

The authors present a genomic analysis of Anguilla eels, connecting evidence from contemporary examples of hybridization with historical introgression, despite significant divergence time estimates between the species. The role of hybridization and introgression to the topic of speciation is of keen importance, and rarely are contemporary instances of hybridization considered alongside robust estimates of historical introgression.

To determine contemporary rates of hybridization, the authors used genomic and morphological analyses to quantify different hybrid classes. Their genomic data included a mix of reduced representation sequencing, as well as whole genome resequencing. They were able to successfully identify several hybrids, and this made up a surprising portion of the sampled individuals. To estimate historical level of possible introgression, the authors used a range of robust phylogenomic tools. Fairly convincingly showing that variation in the topology amongst these trees is the result of historical introgression.

We thank Reviewer 3 for reviewing our manuscript and for considering our analyses robust and convincing.

I found the writing clear and the flow of the manuscript good, although sometimes ending up further into the weeds than I would expect in broad journal like this (e.g. lines 150-191). While I do have some suggested improvements, I think the paper will be of general interest to the readers of Nature Communications and am supportive of publication.

We are glad that Reviewer 3 finds our study of general interest and thank Reviewer 3 for this statement of support. In our revision, we have shortened the most technical sections, like the section on lines 150-191 of the original manuscript (now on lines 144-163) substantially to facilitate reading for a general audience.

Even more importantly, I might request the authors make the figures for some of my own papers—what a breath of fresh air it is to see clear, informative, and thoughtful figures, with a very appropriate use of color (i.e. beautiful evidence!).

We are thrilled that our care for aesthetic figure design is appreciated.

Specific Comments:

Figure 1a – Regarding the positional information, it seems odd to just use the GBIF data, and not include your own collection information. For example, the SAW site obviously has lots of obscure, but that wasn't included in the GBIF dataset, so doesn't show up on the map. I might recommend combining the datasets for the locality information?

We followed this suggestion to combine distribution information from GBIF and our own sampling. Specifically, we corrected the absence of *A. interioris* from Java and the absence of *A. obscura* from Samoa. We updated the legend of Figure 1 accordingly and also adjusted Fig. 3c.

Line 77-78 – Given the overlap in morphospace, it is somewhat difficult to see “intermediate” individuals here? Might be worth revising this sentence.

We agree and rephrased the sentence as “...showed species-specific clusters even though the clusters for *A. marmorata* and *A. megastoma* were not clearly separated from each other (Figure 1b, Supplementary Figure 3).” (lines 74-75).

Figure 2e-f – I guess here I am a bit surprised that the multivariate analysis is showing morphology, and not the genomic analysis, give the genomic analysis has more discriminatory power to cluster the groups.

We had shown plots of morphological variation in Figure 2e-h to demonstrate that hybrids are not just intermediate between parental species genetically (as shown with ancestry painting in the same figure) but also morphologically, and also to illustrate the way in which we calculated $f_{m,morphology}$. However, we agree that the results of the morphological analysis are less directly relevant to the question of hybridization; therefore we have shifted panels e-h from Figure 2 to the Supplementary Information (as a new Supplementary Figure 16), and replaced them in Figure 2 with similar plots showing genomic variation. We also changed the order of panels in Figure 2 and adjusted the figure legend accordingly.

Figure 2i – I appreciate the ancestry painting and the multivariate genomic analysis. I wonder if the authors had also considered using “triangle plot” analyses (e.g. DOI: <https://doi.org/10.1098/rspb.2017.2081>) to identify different hybrid classes?

We were not aware of the study by Pulido-Santacruz et al. but agree that the triangle plots presented in that study are a useful way of illustrating genomic variation among hybridizing species. We have now included a triangle plot in our revised manuscript, but to avoid confusion with the other triangular plot that we use to visualize likelihood support from RAD loci for different topologies (Fig. 3a), we placed the new triangle plot in the Supplementary Information, as Supplementary Figure 15. We refer to it in the Results section “High frequency of contemporary hybridization”.

Lines 162 – 191 – I guess I am not overly surprised by the fact that the putative hybrids are intermediate in size (as would be predicted from a multi-locus quantitative trait, and has been demonstrated quite a few times before in hybrid zones). I also think the transgressive piece here, while interesting, seems somewhat tangential to your main question. Therefore, I might suggest you reduce the emphasis in the text on this morphological analysis, at least here.

We had included the morphological analysis to provide context but agree that its results are not essential for our main question. We therefore followed this suggestion of Reviewer 3 and

have now shortened the description of the results of the morphological analysis in the Results section “High frequency of contemporary hybridization” substantially (lines 144-163).

Figures 3c and d – I was always told not to include a regression line if the relationship was not significant...

We have removed the regression lines for the two non-significant relationships.

Lines 301-302 – On my first reading, this made me think you were suggesting that the production of adult F1 hybrids don't also speak to the possible post-mating barriers. I appreciate the point following from your question about whether barriers that act after the F1 are very important, but I would rephrase this slightly.

We thank Reviewer 3 for pointing out this ambiguity and have now rephrased the sentence to clarify that the postzygotic barrier that we had in mind is a reduction of the fertility of F1 hybrids (lines 319-323).

Reviewers' Comments:

Reviewer #1:

Remarks to the Author:

I thank the authors for their clear and comprehensive replies to the issues I raised. I am satisfied with these answers and the changes they made to the manuscript.

In particular, of course, I am pleased with the inclusion of the extensive new analysis on putative cyto-nuclear incompatibilities. Although the set of identified mechanisms is likely just the tip of the iceberg, the highlighted results are quite compelling, especially since they are consistent with previous studies on different species in the genus *Anguilla*. I am slightly surprised - in a good way! - by such robust results from heterogeneous draft genome assemblies.

I previously commented that this is a convincing study from a population biological perspective, but that 'the responsible mechanisms remain less clear.' With these new analyses, that minor weakness has been addressed. This more complete story will be of great interest to a wide audience, and I recommend Nature Communications accepts it for publication.

Christiaan Henkel

Reviewer #3:

Remarks to the Author:

I have read the revision and the response to reviewers. I was Reviewer #3 on the original submission and believe that the authors have adequately addressed the concerns.

My only issue now is the significant addition of a new analysis quantifying the potential for mitochondrial-nuclear incompatibilities, motivated by the observation of a strong discordance between the two in one of the species pairs. Part of this new analysis is in response to a comment by Reviewer 1, who suggested possibly trying to identify the mechanistic bases for this discordance ... although I think the authors have taken this a bit far in the current version.

My goal with peer-review is not to be a "moving target" during revisions, and I appreciate the authors have added these additional analyses in good faith as a response to previous reviews. That said, I think the analyses go much too far in terms of their ability of identifying mito-nuclear incompatibilities and any mechanistic interpretation. I think the finding of the small inversion is interesting, and the increase in mitochondrial gene non-synonymous substitutions is intriguing. However, these observations seem far from the kind of evidence necessary to generate the specific claims of incompatibilities (i.e. the speculation that forms the basis of lines 286-303). Moreover, the connection between how the differences in the genes *muhc4* and *ttna* are linked to the mitochondrial differences is supported by the fact these gene products are "powered by hydrolysis of ATP". This appears to be a rather tenuous this connection, given the ubiquity of ATP for a vast range of other cellular processes. Given these concerns, I think the evidence is currently much too indirect to support these interpretations.

In the first version of the paper, I found the paper was very even-handed and fair with its interpretations, and the results interesting and of broad interest. Unfortunately, in this version, the addition of lines 202-303 and Figure 4, in my opinion, go beyond the realm of appropriate speculation at this stage. These additional analyses also do not seem to fit as well with the rest of the story in the

manuscript. While interesting patterns, I would strongly recommend at least moving most of this to the supplemental information.

Response to comments by Reviewer 1

I thank the authors for their clear and comprehensive replies to the issues I raised. I am satisfied with these answers and the changes they made to the manuscript.

*In particular, of course, I am pleased with the inclusion of the extensive new analysis on putative cyto-nuclear incompatibilities. Although the set of identified mechanisms is likely just the tip of the iceberg, the highlighted results are quite compelling, especially since they are consistent with previous studies on different species in the genus *Anguilla*. I am slightly surprised - in a good way! - by such robust results from heterogeneous draft genome assemblies.*

I previously commented that this is a convincing study from a population biological perspective, but that 'the responsible mechanisms remain less clear.' With these new analyses, that minor weakness has been addressed. This more complete story will be of great interest to a wide audience, and I recommend Nature Communications accepts it for publication.

Christiaan Henkel

We thank Dr. Henkel for his constructive and thorough review, including the valuable suggestion to explore the underlying causes of cyto-nuclear incompatibility, and we are very pleased that he now recommends our study for publication.

Response to comments by Reviewer 3

I have read the revision and the response to reviewers. I was Reviewer #3 on the original submission and believe that the authors have adequately addressed the concerns.

My only issue now is the significant addition of a new analysis quantifying the potential for mitochondrial-nuclear incompatibilities, motivated by the observation of a strong discordance between the two in one of the species pairs. Part of this new analysis is in response to a comment by Reviewer 1, who suggested possibly trying to identify the mechanistic bases for this discordance ... although I think the authors have taken this a bit far in the current version.

*My goal with peer-review is not to be a "moving target" during revisions, and I appreciate the authors have added these additional analyses in good faith as a response to previous reviews. That said, I think the analyses go much too far in terms of their ability of identifying mito-nuclear incompatibilities and any mechanistic interpretation. I think the finding of the small inversion is interesting, and the increase in mitochondrial gene non-synonymous substitutions is intriguing. However, these observations seem far from the kind of evidence necessary to generate the specific claims of incompatibilities (i.e. the speculation that forms the basis of lines 286-303). Moreover, the connection between how the differences in the genes *muhc4* and *ttna* are linked to the mitochondrial differences is supported by the fact these gene products are "powered by hydrolysis of ATP". This appears to be a rather tenuous this connection, given the ubiquity of ATP for a vast range of other cellular processes. Given these concerns, I think the evidence is currently much too indirect to support these interpretations.*

In the first version of the paper, I found the paper was very even-handed and fair with its interpretations, and the results interesting and of broad interest. Unfortunately, in this version, the addition of lines 202-303 and Figure 4, in my opinion, go beyond the realm of appropriate speculation at this stage. These additional analyses also do not seem to fit as well with the rest of the story in the manuscript. While interesting patterns, I would strongly recommend at least moving most of this to the supplemental information.

We thank Reviewer 3 for this evaluation of our revised manuscript. We have followed the suggestion of the Editors to keep the interpretation of the observed signals of cytonuclear incompatibilities within the main text with a more careful presentation. We have thus adjusted the Abstract, the end of the Introduction, and the corresponding Results section to clarify that our results are suggestive but not definite evidence of cytonuclear incompatibilities.